# REINFORCEMENT LEARNING VIA LAZY-AGENT FOR ENVIRONMENTS WITH RANDOM DELAYS

## ABSTRACT

Real-world reinforcement learning applications are often hampered by delayed feedback from environments, which violates the fundamental assumption of the Markovian property and introduces significant challenges. While numerous methods have been proposed for handling environments with constant delays, those with random delays remain largely unexplored owing to their inherent complexity and variability. In this study, we explored environments with random delays and proposed a novel strategy to transform them into their equivalent constant-delay counterparts by introducing a simple agent called the *lazy-agent*. This approach naturally overcomes the challenges posed by the variability of random delays, enabling the application of state-of-the-art methods, originally designed for handling constant delays, to random-delay environments without any modification. Empirical results demonstrate that the lazy-agent-based algorithm significantly outperformed other baselines in terms of asymptotic performance and sample efficiency in random-delay environments.

## 1 INTRODUCTION

Reinforcement learning (RL) has made remarkable progress in various domains, from gaming (Mnih et al., 2013; Silver et al., 2016) to robotic control systems (Haarnoja et al., 2018; Kalashnikov et al., 2018). However, real-world applications of RL often face challenges due to delays, which can take diverse forms such as latency in communication systems, delays in processing sensory data, or response delays from actuators. These delays can significantly degrade the performance of RL agents and may even cause instability in dynamic systems (Hwangbo et al., 2017; Mahmood et al., 2018).

While numerous methods have been proposed to address the challenges posed by delays within the RL framework, these efforts primarily focus on the unrealistic assumption of constant delays (Chen et al., 2021; Derman et al., 2021; Liotet et al., 2022; Kim et al., 2023; Wu et al., 2024), leaving random delays relatively unexplored owing to their inherent complexity and variability. However, in real-world, randomly varying delays present a more realistic challenge, exemplified by communication systems where diverse routing paths and the physical properties of the network can result in asynchronous data arrivals (Ge et al., 2013).

In this study, we explore environments with random delays and establish a connection to environments with constant delays by introducing a simple agent called the *lazy-agent*. Specifically, we demonstrate that random-delay environments can be straightforwardly transformed into their equivalent constant-delay counterparts using lazy-agents, enabling state-of-the-art constant-delay approaches to be seamlessly applied to random-delay environments without any modification. We train lazy-agents within the belief projection-based $Q$-learning (BPQL) framework (Kim et al., 2023), termed *lazy-BPQL*, to leverage its advantages in training agents in delayed environments. The efficacy of the proposed lazy-BPQL was evaluated on popular continuous control tasks in the MuJoCo benchmark (Todorov et al., 2012). Empirical results demonstrate that lazy-BPQL outperformed other baseline algorithms in terms of asymptotic performance and sample efficiency in random-delay environments, achieving performance comparable to agents trained in constant-delay environments.

## 2 BACKGROUNDS

### 2.1 STANDARD REINFORCEMENT LEARNING

A (*delay-free*) Markov decision process (MDP) (Bellman, 1957) can be defined with a five-tuples $(\mathcal{S}, \mathcal{A}, \mathcal{P}, \mathcal{R}, \gamma)$, where $\mathcal{S}$ is the state space, and $\mathcal{A}$ is the action space, $\mathcal{P} : \mathcal{S} \times \mathcal{A} \times \mathcal{S} \to [0, 1]$ is the transition kernel, $\mathcal{R} : \mathcal{S} \times \mathcal{A} \to \mathbb{R}$ is the reward function, and $\gamma \in (0, 1)$ is a discount factor. Additionally, the policy $\pi : \mathcal{S} \times \mathcal{A} \to [0, 1]$ maps the state-to-action distribution.

Under this definition, at each discrete time $t$, an RL agent observes state $s_t$, makes a decision $a_t$ based on a policy $\pi$, receives a reward $r_t$ with respect to the action taken, and then observes the next state $s_{t+1}$ from the environment. It repeats this process to find an optimal policy $\pi^*$ that maximizes the expected discounted cumulative rewards, which is given as:

$$\pi^* := \arg\max_{\pi} \; \mathbb{E}\left[\sum_{k=0}^{H-1} \gamma^k r_{k+1} | \pi, \rho_0\right] = \arg\max_{\pi} \; \mathbb{E}\left[G_0 | \pi, \rho_0\right], \tag{1}$$

where $\rho_0$ denotes the initial state distribution and $G_0$ is the discounted cumulative rewards starting from the initial state over a finite or infinite-horizon $H$ under the policy $\pi$. Additionally, the values of states and actions at time $t$ are defined as:

$$V^\pi(s) = \mathbb{E}\left[\sum_{k=0}^{H-1} \gamma^k r_{t+k+1} | \mathcal{S}_t = s, \pi\right], \; Q^\pi(s, a) = \mathbb{E}\left[\sum_{k=0}^{H-1} \gamma^k r_{t+k+1} | \mathcal{S}_t = s, \mathcal{A}_t = a, \pi\right], \tag{2}$$

where $V^\pi(s)$ denotes the expected discounted cumulative rewards starting from state $s$ under the policy $\pi$, and $Q^\pi(s, a)$ represents the expected discounted cumulative rewards starting from state $s$, taking action $a$, and then following the policy $\pi$.

Note that the dynamics governing MDPs assume the Markovian property, which indicates that the complete probability distribution in the dynamics can be fully determined by the present state and action, independent of their histories. However, this fundamental assumption can be violated by delayed feedback from the environment, leading to partially observable MDPs (Monahan, 1982), where the agent's current state and action cannot capture sufficient information needed for timely decision-making. This can significantly degrade the performance of RL agents or even lead to complete failure in learning (Singh et al., 1994).

### 2.2 DELAYED REINFORCEMENT LEARNING

In MDP with delays, referred to as *delayed* MDP, the state of the environment may not be observed by the agent immediately (observation delay). The effect of the action applied to the environment may also be delayed (action delay). Additionally, the reward generated by the action taken may not reach the agent immediately (reward delay). These delays force the agent to make decisions based on outdated information, prevent timely and appropriate actions, or cause the agent to receive rewards that do not correspond to the actions taken, disrupting the learning process.

Delayed MDPs are typically categorized into *constant-delay* MDPs (CDMDPs), where feedback is delayed by a fixed number of time-steps; and *random-delay* MDPs (RDMDPs), where the number of delayed time-steps varies randomly. For example, in the case of random observation delay, the state $s_t$ may be delayed by four time-steps and observed by the agent at time $t + 4$, whereas the next state $s_{t+1}$ may be delayed by only one time-step and observed at time $t + 2$. Note that the subscript $t$ explicitly indicates the times when states are generated by the actions applied to the environment.

In this study, we focus on randomly delayed observations under the assumption that the agent utilizes them for decision-making *in order*. This implies that any observed state can be used by the agent for decision-making only after all previously generated states have been both observed and utilized to ensure no state is omitted from the decision-making process. In the presence of randomness in observation delays, multiple states may become observable simultaneously, and their order may even be scrambled. To reduce confusion about the timing of state observations, we distinguish between a state being *observed* and *used*, under the aforementioned assumption of *ordering*, as follows:

**Definition 2.1.** A state is considered *observed* when the information about the state of the environment reaches the agent. A state is considered *used* when the agent utilizes the observed state information to make a decision by feeding it into the policy.

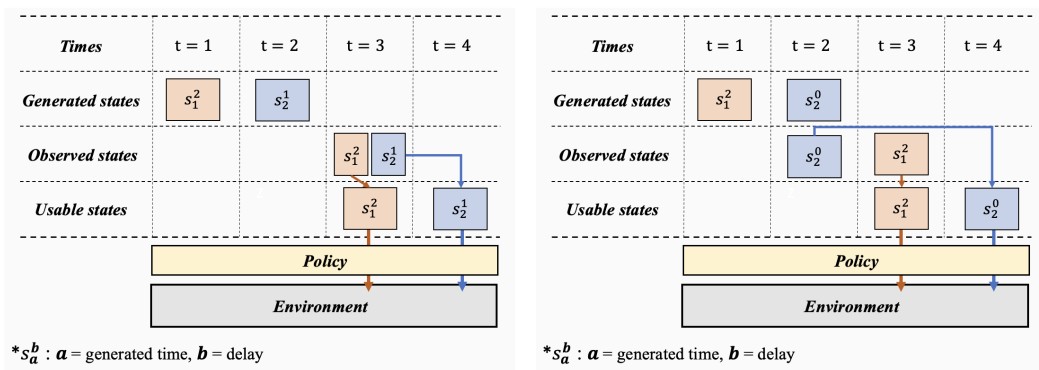

(a) When states are observed simultaneously     (b) When the order of state observation is scrambled

Figure 1: Visual examples illustrating cases where (a) states are observed simultaneously and (b) the order of state observation is scrambled. In both scenarios, an observed state is not available for use if a previously generated state has not yet been used in the decision-making process.

Suppose the state $s_1$ has an observation delay of 2, and the subsequent state $s_2$ has an observation delay of 1. These delayed states are *observed* by the agent at time 3 simultaneously. However, despite being observed, $s_2$ is not immediately *usable* at this time because the preceding state has not yet been used in the decision-making process. Thus, the agent uses the observed states in sequence to determine its actions: $s_1$ is used at time 3 and $s_2$ is used at time 4 (Fig. 1(a)). Throughout this study, the term 'observe' is utilized when a strict distinction between the two is unnecessary, specifically in cases of constant delays.

## 3 AUGMENTATION-BASED APPROACH

An augmentation-based approach is often preferred in delayed MDPs, as it retrieves the Markovian property and offers advantages for agents learning policies through conventional RL algorithms in such environments (Liotet et al., 2022; Kim et al., 2023; Wu et al., 2024). As demonstrated by Altman & Nain (1992); Katsikopoulos & Engelbrecht (2003), delayed MDPs can be reduced to equivalent MDPs without delays through this approach, known as *regular* MDPs, where the resulting optimal policies are optimal in the original delayed MDPs (Bander & White III, 1999; Katsikopoulos & Engelbrecht, 2003). The augmentation-based approach involves state augmentation, where the state is concatenated with additional delay-related information, similar to methods employed in conventional control theory (Kwon & Pearson, 1980; Park et al., 2008). In this section, we examine two types of delayed MDPs: CDMDPs and RDMDPs.

### 3.1 CONSTANT-DELAY MDPS

CDMDPs can be defined as a six-tuple $(\mathcal{S}, \mathcal{A}, \mathcal{P}, \mathcal{R}, \gamma, o)$, where $o \in \mathbb{N}$ is a constant variable representing the observation delay. As demonstrated by Katsikopoulos & Engelbrecht (2003), it is reducible to regular MDPs $(\mathcal{X}_o, \mathcal{A}, \mathcal{P}, \bar{\mathcal{R}}, \gamma)$, where $\mathcal{X}_o = \mathcal{S} \times \mathcal{A}^o$ is the augmented state space with $\mathcal{A}^o$ being the Cartesian product of $\mathcal{A}$ with itself for $o$ times, and $\bar{\mathcal{R}} : \mathcal{X}_o \times \mathcal{A} \to \mathbb{R}$ is the reward function with respect to the augmented state space, termed the augmented reward function. Finally, the augmented state-based policy $\bar{\pi} : \mathcal{X}_o \times \mathcal{A} \to [0, 1]$ maps augmented state-to-action distribution.

To be specific, the augmented state at time $t$ is defined as:

$$x_t = (s_{t-o}, a_{t-o}, a_{t-o+1}, ..., a_{t-1}), \quad \forall t > o, \tag{3}$$

where $s_{t-o}$ is the most recently *observed* state and $(a_{t-o}, ..., a_{t-1})$ is the history of actions taken since $s_{t-o}$ was generated. The agent implicitly estimates unobserved state $s_t$ based on the augmented state $x_t$ and selects action $a_t$ accordingly. Note that since $s_t$ is not explicitly known at time $t$, the augmented reward corresponding to the action $a_t$ becomes a random variable that has to be determined based on the conditional expectation, which is given as $\bar{R}(x_t, a_t) := \mathbb{E}_{\mathbb{P}(s_t|x_t)}[R(s_t, a_t)]$.

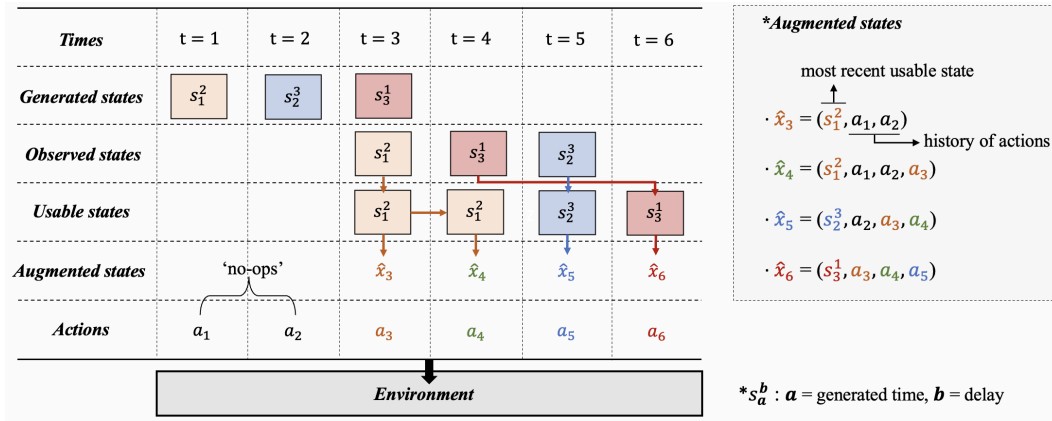

Figure 2: A visual example of decision-making processes in environments with random delays. At time 3, state $s_1^2$ is observed, and the agent immediately uses this observed state to make a decision. At time 4, state $s_3^1$ is observed, whereas the preceding state $s_2^3$ remains unobserved. Consequently, $s_3^1$ cannot be used at this time because the previously generated state $s_2^3$ has not yet been observed and used. Therefore, the agent continues to use $s_1^2$ until $s_2^3$ is observed. Eventually, $s_2^3$ is observed at time 5, and the observed states are then used in the correct order. Note that the states are reformulated as augmented states before being fed into the policy, which subsequently determines the appropriate actions.

## 3.2 RANDOM-DELAY MDPS

Using similar arguments as in Katsikopoulos & Engelbrecht (2003), RDMDPs can be defined as an eight-tuple $(\mathcal{S}, \mathcal{A}, \mathcal{P}, \mathcal{R}, \gamma, O, q_o, \tau)$, where $O \in \{0\} \cup \mathbb{N}$ is a random variable representing varying observation delay, assumed to be sampled from an arbitrary discrete distribution $q_o$ with support on $\{0, 1, ..., o_{\max}\}$, where $o_{\max}$ denotes a maximum delay in time-steps, and $\tau : \mathcal{S} \to \mathbb{N}$ is a time-related function that maps states to times at which they are used to make decisions for the first time. Under this definition, we can define the states with observation delays and their corresponding augmented states as follows:

**Definition 3.1.** Let $s_n^{o_n}$ be a state with an observation delay of $o_n$, where the subscript $n > 0$ denotes the time at which the state was generated. Given that $s_n^{o_n}$ is the most recent *usable* state at time $t$, the augmented state at time $t$ is defined as:

$$\hat{x}_t = (s_n^{o_n}, a_n, a_{n+1}, ..., a_{t-1}), \quad \text{for } \tau(s_n^{o_n}) \le t \le n + o_{\max}, \tag{4}$$

where the sequence $(a_n, a_{n+1}, ..., a_{t-1})$ represents the history of actions taken since $s_n^{o_n}$ was generated, and $\tau(s_n^{o_n}) \in \{n + o_n, ..., t\}$ denotes the time when $s_n^{o_n}$ is used for decision-making for the first time. Based on the augmented state $\hat{x}_t$, the agent implicitly estimates the unobserved state $s_t$ and selects action $a_t$ accordingly. The notation $\hat{\cdot}$ indicates the augmented state defined in RDMDPs.

Given that $\tau(s_n^{o_n}) = t$ and $s_{n+1} \sim \mathcal{P}(\cdot|s_n, a_n)$, the usability of the next state $s_{n+1}^{o_{n+1}}$ at time $t + 1$ depends on its delay $o_{n+1}$. Consequently, the next augmented state $\hat{x}_{t+1}$ is defined as:

$$\hat{x}_{t+1} = \begin{cases} (s_{n+1}^{o_{n+1}}, a_{n+1}, a_{n+2}, ..., a_t) & \text{with prob. of } m(o_{n+1}), \\ (s_n^{o_n}, a_n, a_{n+1}, ..., a_{t-1}, a_t) & \text{with prob. of } 1 - m(o_{n+1}), \end{cases} \tag{5}$$

$$\text{where } m(o_{n+1}) = Pr\big(o_{n+1} \in \{0, 1, ..., \tau(s_n^{o_n}) - n\}\big).$$

Note that state $s_n^{o_n}$ continues to be used if the next state $s_{n+1}^{o_{n+1}}$ is not available for use at this time. It remains in use until $s_{n+1}^{o_{n+1}}$ becomes usable, at which point the agent switches to use $s_{n+1}^{o_{n+1}}$ to construct the augmented state (Fig. 2).

It is important to note that the dimension of the augmented state $\hat{x}_{t+1}$ in equation 5 either remains constant ($|\hat{x}_{t+1}| = |\hat{x}_t|$) or increases by 1 ($|\hat{x}_{t+1}| = |\hat{x}_t| + 1$). This implies that its dimension will eventually reach infinity in infinite-horizon MDPs without assuming a bounded maximum delay. Under the assumption of bounded maximum delays, Katsikopoulos & Engelbrecht (2003) proposed

a method called *freeze*. In this approach, the agent performs no actions ('no-ops') until the current state $s$ is observed whenever the dimension of the augmented state reaches its maximum tolerable limit. Once the state $s$ is observed, the augmented state is reset to $\hat{x} = (s)$, and the decision-making process resumes. However, this approach is known to be highly task-dependent, as the agent ignores environmental changes during the inactive periods, potentially leading to suboptimal policies.

# 4    BELIEF PROJECTION-BASED Q-LEARNING

While the augmentation-based approach provides a foundation for training agents with conventional RL algorithms in delayed MDPs, it has a known limitation: the state space grows exponentially as the number of delayed time-steps increases, resulting in sample inefficiency and slow convergence. This is called the state-space explosion issue, which makes the augmentation-based approach unfavorable for environments with long delays (Derman et al., 2021; Kim et al., 2023).

To mitigate this issue, belief projection-based $\mathcal{Q}$-learning (BPQL), a model-free actor-critic framework designed to handle constant-delay environments, was proposed by Kim et al. (2023). It exhibited remarkable performance with simple modifications to the conventional augmentation-based approach, effectively alleviating the state-space explosion issue.

## 4.1    ALTERNATIVE REPRESENTATIONS FOR AUGMENTATION-BASED VALUES

First, a modified Bellman operator $\bar{\mathcal{T}}$, termed *delay Bellman operator*, was introduced to evaluate the values with respect to the augmented state space, which is given as:

$$\bar{\mathcal{T}}^{\bar{\pi}} \bar{V}^{\bar{\pi}}(x_t) \mapsto \mathbb{E}_{a_t \sim \bar{\pi}(\cdot|x_t)} \left[ \mathbb{E}_{\mathbb{P}(s_t|x_t)} \left[ R(s_t, a_t) \right] + \gamma \mathbb{E}_{x_{t+1} \sim \bar{\mathcal{P}}(\cdot|x_t, a_t)} \left[ \bar{V}^{\bar{\pi}}(x_{t+1}) \right] \right], \quad \forall t > o, \quad (6)$$

where $\bar{\pi}$ is the augmented state-based policy that receives augmented states as input, $\bar{V}^{\bar{\pi}}$ is the augmented state-based value representing the values of augmented states under the policy $\bar{\pi}$, $o$ is a constant observation delay, and $\bar{\mathcal{P}} : \mathcal{X}_o \times \mathcal{A} \times \mathcal{X}_o \to [0, 1]$ is the transition kernel defined with respect to the augmented state space. By repeatedly applying the delay Bellman operator, $\bar{V}^{\bar{\pi}}$ converges, and then $\bar{\pi}$ is improved using conventional policy improvement methods. Similarly, the augmented state-based $Q$-value, $\bar{Q}^{\bar{\pi}}$, representing the values of augmented states for the given actions under the policy $\bar{\pi}$, can also be employed.

To mitigate the state-space explosion issue, the alternative representations for $\bar{V}$ and $\bar{Q}$, referred to as *beta value* and *beta Q-value* ($V_\beta$ and $Q_\beta$), are introduced. These values are evaluated with respect to the original state space rather than the augmented one, thereby naturally alleviating the state-space explosion issue. Beta-based values can be used as estimators for the augmentation-based values, which are given as:

$$\bar{V}^{\bar{\pi}}(x_t) = \mathbb{E}_{\mathbb{P}(s_t|x_t)} \left[ V_\beta^{\bar{\pi}}(s_t) \right] + \Delta_{\text{residual}}^{\bar{\pi}}(x_t) \tag{7}$$

$$\bar{Q}^{\bar{\pi}}(x_t, a_t) = \mathbb{E}_{\mathbb{P}(s_t|x_t)} \left[ Q_\beta^{\bar{\pi}}(s_t, a_t) \right] + \delta_{\text{residual}}^{\bar{\pi}}(x_t, a_t), \tag{8}$$

where $\Delta_{\text{residual}}^{\bar{\pi}}$ and $\delta_{\text{residual}}^{\bar{\pi}}$ represent the projection residuals.

In large or continuous spaces, a practical sampling-based reinforcement learning algorithm can be employed, where the beta $Q$-value and augmented state-based policy are parameterized by $\theta$ (beta critic) and $\phi$ (actor), respectively. The beta critic and actor are then trained by iteratively minimizing the following objective functions:

$$\mathcal{J}_{Q_\beta}(\theta) = \mathbb{E}_{(s_t, a_t, r_t, s_{t+1}, x_{t+1}) \sim \mathcal{D}} \left[ \frac{1}{2} \left( Q_{\theta, \beta}^{\bar{\pi}}(s_t, a_t) - R(s_t, a_t) \right. \right.$$

$$\left. \left. - \gamma \mathbb{E}_{a_{t+1} \sim \bar{\pi}_\phi(\cdot|x_{t+1})} \left[ Q_{\tilde{\theta}, \beta}^{\bar{\pi}}(s_{t+1}, a_{t+1}) - \alpha \log \bar{\pi}_\phi(a_{t+1}|x_{t+1}) \right] \right)^2 \right], \tag{9}$$

$$\mathcal{J}_{\bar{\pi}}(\phi) = \mathbb{E}_{(s_t, x_t) \sim \mathcal{D}} \left[ \mathbb{E}_{a_t \sim \bar{\pi}_\phi(\cdot|x_t)} \left[ \alpha \log \bar{\pi}_\phi(a_t|x_t) - Q_{\theta, \beta}^{\bar{\pi}}(s_t, a_t) \right] \right], \tag{10}$$

where $r_t = R(s_t, a_t)$, $\mathcal{D}$ represents a replay buffer (Mnih et al., 2013), $\alpha$ is a temperature parameter (Haarnoja et al., 2018), and $\tilde{\theta}$ are the parameters of the target beta critic (Fujimoto et al., 2018).

Consequently, it demonstrated remarkable performance in constant-delay environments. However, since it was specifically designed to handle constant delays, it is not applicable to random-delay environments, which are the focus of our study. As discussed earlier, we demonstrate that conventional constant-delay approaches can be naturally applied to random-delay environments by establishing a connection between RDMDPs and CDMDPs, thereby facilitating the application of the BPQL framework in our study.

## 5 BRIDGING RDMDPS TO CDMDPS

In this section, we demonstrate that RDMDPs can be transformed into their equivalent CDMDPs by introducing a simple agent called lazy-agent, allowing state-of-the-art constant-delay approaches to be seamlessly extended to random-delay environments.

### 5.1 LAZY-AGENT

To address the challenges caused by variability in random delays, we let the agent assume that all states are delayed by the maximum number of time-steps. The agent then uses the observed states in decision-making processes at their maximum delayed times, that is, $\tau(s_n^{o_n}) = n + o_{\max}, \forall n > 0$. In this scheme, each state is consistently used in sequence exactly $o_{\max}$ time-steps after being generated, regardless of its actual delay.

Suppose the state $s_1^{o_1}$ is observed at time $1 + o_1$. Since the agent assumes that all states are delayed by $o_{\max}$ time-steps, it uses the observed state at time $1 + o_{\max}$ irrespective of its actual delay $o_1$. Similarly, the subsequent state $s_2^{o_2}$ is observed at time $2 + o_2$; however, the agent uses it at time $2 + o_{\max}$. In short, the agent uses observed states at their maximum delayed times, regardless of their actual delays. This implies that the exact delays for each state may remain unknown to the agent, except for the maximum delay. Consequently, this approach effectively circumvents the challenges associated with variability in random delays. We refer to this agent as a *lazy-agent*. A visual representation is provided in Appendix E.

Formally, the augmented state $\hat{x}_t$ in equation 4 is redefined for the lazy-agent as:

$$\hat{x}_t = (s_n^{o_n}, a_n, a_{n+1}, ..., a_{t-1}), \quad \text{for } t = n + o_{\max}, \ \forall n > 0, \tag{11}$$

where $\tau(s_n^{o_n}) = n + o_{\max}$, resulting in the probability $m(o_{n+1})$ in equation 5 becoming 1. Consequently, given $s_{n+1} \sim \mathcal{P}(\cdot|s_n, a_n)$, the next augmented state $\hat{x}_{t+1}$ in equation 5 is redefined as:

$$\hat{x}_{t+1} = (s_{n+1}^{o_{n+1}}, a_{n+1}, a_{n+2}, ..., a_t), \quad \text{for } t = n + o_{\max}, \ \forall n > 0, \tag{12}$$

irrespective of its delay $o_{n+1}$.

Note that equation 11 and equation 12 align with the formulations for the augmented states defined in CDMDPs with a constant observation delay of $o_{\max}$. Furthermore, the dimension of the augmented state remains constant at $(o_{\max} + 1)$ at all times, addressing the issue of an exploding augmented state dimension in infinite-horizon MDPs.

Consequently, RDMDPs now become equivalent to CDMDPs with a constant delay of $o_{\max}$ to the lazy-agents. The resulting CDMDPs can then be further reduced to regular MDPs, allowing the lazy-agents to be trained using conventional RL algorithms. To support our analysis, we present empirical results in Appendix B.2, demonstrating that the performance of lazy-agents trained in random-delay environments is comparable to that of agents trained in constant-delay environments. From these empirical results, we propose the following proposition:

**Proposition 5.1.** *Under the assumption of bounded observation delay and ordering, the RDMDPs* $(\mathcal{S}, \mathcal{A}, \mathcal{P}, \mathcal{R}, \gamma, O, q_o, \tau)$ *can be transformed into the equivalent CDMDPs* $(\mathcal{S}, \mathcal{A}, \mathcal{P}, \mathcal{R}, \gamma, o_{max})$ *through the lazy-agent, where $o_{max}$ denotes the maximum delay in RDMDPs.*

*Proof sketch.* We begin by introducing the lazy-agent into RDMDPs, in which the agent assumes that all states are delayed by the maximum number of time-steps ($o_{\max}$) and uses the observed states for decision-making at their maximum delayed times, that is, $\tau(s_n^{o_n}) = n + o_{\max}, \forall n > 0$.

In this setting, the augmented state $\hat{x}_t$ in equation 4 is redefined as:

$$\hat{x}_t = (s_n^{o_n}, a_n, a_{n+1}, ..., a_{t-1}), \quad \text{for } t = n + o_{\max}, \ \forall n > 0, \tag{13}$$

indicating that $s_n^{o_n}$ is *used* for decision-making only at time $n + o_{\max}$, regardless of its actual delay. Under this assumption, the probability $m(o_{n+1})$ in equation 5 becomes:

$$m(o_{n+1}) = Pr\big(o_{n+1} \in \{0, 1, ..., \tau(s_n^{o_n}) - n\}\big) = Pr\big(o_{n+1} \in \{0, 1, ..., o_{\max}\}\big) = 1. \quad (14)$$

Consequently, the next augmented state $\hat{x}_{t+1}$ in equation 5 is redefined as:

$$\hat{x}_{t+1} = (s_{n+1}^{o_{n+1}}, a_{n+1}, a_{n+2}, ..., a_t), \quad \text{for } t = n + o_{\max}, \ \forall n > 0, \quad (15)$$

where $s_{n+1} \sim \mathcal{P}(\cdot|s_n, a_n)$. Evidently, these formulations for the augmented states are equivalent to those defined in CDMDPs with a constant delay of $o_{\max}$. To be more specific, by replacing $n$ with $t - o_{\max}$ in equation 13, we obtain:

$$\hat{x}_t = (s_{t-o_{\max}}^{o_n}, a_{t-o_{\max}}, a_{t-o_{\max}+1}, ..., a_{t-1}), \quad \forall t > o_{\max}, \quad (16)$$

which is exactly equivalent to equation 3 with $o = o_{\max}$, demonstrating that RDMDPs can be transformed into their equivalent CDMDPs through the lazy-agent. This completes the proof. $\qquad \square$

## 5.2 LAZY-BPQL

In the previous section, we demonstrated that RDMDPs can become equivalent to CDMDPs with lazy-agents. However, since the delays in the resulting CDMDPs are determined by the maximum delays in RDMDPs, the lazy-agents often encounter long-delay challenges, particularly the state-space explosion issue. Specifically, the augmented state space in derived CDMDPs would be defined as $\mathcal{X}_{o_{\max}} = \mathcal{S} \times \mathcal{A}^{o_{\max}}$, necessitating numerous samples for the augmented-based values to converge.

To address this issue, we employ lazy-agents within the BPQL framework to leverage its advantage in training agents in constant-delay environments while effectively alleviating the state-space explosion problem. We refer to this approach as *lazy-BPQL*, which is summarized in Algorithm 1.

## 6 EXPERIMENTS

### 6.1 BENCHMARKS AND BASELINE ALGORITHMS

We evaluated our algorithm on popular continuous control tasks in the MuJoCo benchmark by gradually increasing the maximum delay $o_{\max}$ from 5 to 20, to assess its performance and robustness with respect to the degree of randomness in delays. In our experiments, we assumed that random delays are sampled from a discrete uniform distribution. Details of the benchmark environments and experiments are provided in Appendix D.

The following algorithms are included in experiments: normal SAC (Haarnoja et al., 2018), delayed-SAC (Derman et al., 2021; Kim et al., 2023), lazy-BPQL, and DC/AC (Bouteiller et al., 2020). The normal SAC adopts a naive approach that selects actions for currently usable states on a memoryless basis and performs 'no-ops' otherwise, without addressing the violation of the Markovian assumption in delayed MDPs. Delayed-SAC is a variant of delayed-$Q$ (Derman et al., 2021) adapted by Kim et al. (2023) for application in continuous spaces. It employs an approximate forward model to explicitly predict unobserved states, which can be learned from transition samples collected in undelayed environments. With this model, the agent recursively predicts unobserved states through one-step predictions repeated over delayed time-steps and selects actions based on predicted states. Lastly, DC/AC is an improved version of SAC that implements an off-policy multi-step value estimation combined with a partial trajectory resampling method, significantly enhancing sample efficiency and demonstrating notable performance in environments with both constant and random delays.

### 6.2 RESULTS

#### 6.2.1 PERFORMANCE COMPARISON

Table 1 and Fig. 3 show the performance of each algorithm on the MuJoCo tasks. The results indicate that the proposed lazy-BPQL demonstrates remarkable performance across all tasks, from relatively short delays ($o_{\max} = 5$) to long delays ($o_{\max} = 20$). In contrast, normal SAC performs poorly across all tasks, as it trains the agent directly in delayed environments without recovering the Markovian property, resulting in nearly random outcomes. Despite respectable performance in relatively simple

tasks with small spaces, delayed-SAC exhibits unsatisfactory task-dependent performance, possibly due to the accumulation of nonnegligible prediction errors as the complexity of task or $o_{max}$ increases, underscoring the need for a more carefully designed dynamics model. Lastly, while DC/AC performs reasonably well in some tasks with short delays, its performance significantly deteriorates as the degree of randomness in delays increases. To further highlight the performance achieved by lazy-BPQL compared to other baseline algorithms, we report the delay-free normalized scores (Wu et al., 2024) in Fig. 7 and Table 2 in Appendix B.1.

Table 1: Results of the MuJoCo tasks with random delays of $o_{max} \in \{5, 10, 20\}$. Each algorithm was evaluated for one million time-steps over five trials with different seeds. The standard deviations of average returns are denoted by $\pm$, and the best performance is in **bold**. Results for constant delays are in blue. Additionally, *delay-free SAC* serves as the baseline performance in delay-free environments.

| Environment | | Ant-v3 | HalfCheetah-v3 | Hopper-v3 | Walker2d-v3 | Humanoid-v3 | InvertedPendulum-v2 |
|---|---|---|---|---|---|---|---|
| $o_{max}$ | Algorithm | | | | | | |
| $\times$ | Random policy | $-58.7_{\pm4}$ | $-285.01_{\pm3}$ | $18.6_{\pm2}$ | $1.9_{\pm1}$ | $121.9_{\pm2}$ | $5.6_{\pm1}$ |
| | Delay-free SAC | $3279.2_{\pm180}$ | $8608.4_{\pm57}$ | $2435.2_{\pm23}$ | $3305.5_{\pm234}$ | $3228.1_{\pm410}$ | $964.3_{\pm29}$ |
| 5 | Normal SAC | $-76.6_{\pm4}$ | $-279.5_{\pm5}$ | $89.2_{\pm10}$ | $44.7_{\pm21}$ | $403.9_{\pm5}$ | $32.2_{\pm2}$ |
| | DC/AC | $907.5_{\pm90}$ | $2561.8_{\pm92}$ | $1931.6_{\pm192}$ | $2079.3_{\pm122}$ | $2798.4_{\pm452}$ | $854.7_{\pm30}$ |
| | Delayed-SAC | $986.4_{\pm128}$ | $4569.4_{\pm88}$ | $\mathbf{2200.4}_{\pm190}$ | $1910.1_{\pm247}$ | $418.9_{\pm126}$ | $\mathbf{964.2}_{\pm15}$ |
| | **Lazy-BPQL** (proposed) | $\mathbf{3679.8}_{\pm167}$ | $\mathbf{5583.9}_{\pm169}$ | $2174.1_{\pm155}$ | $\mathbf{2843.2}_{\pm272}$ | $\mathbf{3157.7}_{\pm292}$ | $958.8_{\pm14}$ |
| | BPQL (constant-delay) | $3761.9_{\pm112}$ | $5212.7_{\pm41}$ | $2136.3_{\pm158}$ | $2577.4_{\pm157}$ | $3194.9_{\pm374}$ | $955.9_{\pm28}$ |
| 10 | Normal SAC | $-84.6_{\pm9}$ | $-278.6_{\pm6}$ | $28.1_{\pm6}$ | $40.9_{\pm4}$ | $354.5_{\pm12}$ | $31.3_{\pm1}$ |
| | DC/AC | $342.9_{\pm34}$ | $1824.5_{\pm111}$ | $1262.4_{\pm261}$ | $1492.5_{\pm133}$ | $1023.8_{\pm359}$ | $4.9_{\pm0}$ |
| | Delayed-SAC | $966.9_{\pm180}$ | $2563.8_{\pm215}$ | $1878.5_{\pm176}$ | $1264.6_{\pm233}$ | $289.6_{\pm108}$ | $\mathbf{947.6}_{\pm36}$ |
| | **Lazy-BPQL** (proposed) | $\mathbf{2744.5}_{\pm112}$ | $\mathbf{4810.1}_{\pm233}$ | $\mathbf{2300.9}_{\pm164}$ | $2122.3_{\pm292}$ | $2820.5_{\pm348}$ | $936.9_{\pm38}$ |
| | BPQL (constant-delay) | $2831.9_{\pm103}$ | $4282.2_{\pm203}$ | $2129.2_{\pm184}$ | $2331.6_{\pm252}$ | $2891.5_{\pm357}$ | $934.7_{\pm20}$ |
| 20 | Normal SAC | $-83.1_{\pm9}$ | $-264.9_{\pm5}$ | $27.5_{\pm5}$ | $64.6_{\pm1}$ | $364.3_{\pm7}$ | $24.3_{\pm0}$ |
| | DC/AC | $258.3_{\pm42}$ | $860.9_{\pm288}$ | $12.8_{\pm6}$ | $-2.9_{\pm5}$ | $237.3_{\pm73}$ | $4.1_{\pm0}$ |
| | Delayed-SAC | $955.7_{\pm110}$ | $1377.8_{\pm140}$ | $1164.1_{\pm278}$ | $811.5_{\pm163}$ | $370.3_{\pm17}$ | $\mathbf{933.5}_{\pm33}$ |
| | **Lazy-BPQL** (proposed) | $\mathbf{1976.5}_{\pm248}$ | $\mathbf{3727.2}_{\pm279}$ | $\mathbf{1346.7}_{\pm245}$ | $\mathbf{1025.7}_{\pm302}$ | $\mathbf{1143.8}_{\pm371}$ | $566.9_{\pm88}$ |
| | BPQL (constant-delay) | $2078.9_{\pm157}$ | $3062.7_{\pm252}$ | $1526.7_{\pm227}$ | $846.7_{\pm443}$ | $1197.7_{\pm457}$ | $608.7_{\pm210}$ |

### 6.2.2 PERFORMANCE IN ENVIRONMENTS WITH CONSTANT DELAYS AND RANDOM DELAYS

To verify whether the proposed lazy-agents perform in random-delay environments as if they were in constant-delay environments, we evaluated the performance of lazy-lazy agents trained in environments with random delays of $o_{max} \in \{5, 10, 20\}$ (lazy-BPQL), and compared it with normal agents trained in environments with constant delays of $o = o_{max}$ (BPQL). The objective was to determine whether these two types of agents demonstrate similar performance. Table 1 presents the results for the MuJoCo tasks. The empirical findings confirm that both types of agents exhibited **almost identical** performance across all evaluated tasks, which strongly supports our argument that random-delay environments can be transformed into their equivalent constant-delay counterparts with the use of lazy-agents. Additional results are provided in Appendix B.2.

### 6.2.3 STATE-SPACE EXPLOSION ISSUE

To highlight the importance of mitigating the state-space explosion issue, we trained lazy-agents solely based on the augmentation-based approach without employing BPQL techniques, which we refer to as *lazy-augmented-SAC*. As presented in Table 5 in Appendix C.1, lazy-BPQL outperforms lazy-augmented-SAC for all evaluated tasks. Note that lazy-augmented-SAC completely failed to learn any useful policy even for tasks with $o_{max} = 5$. These results clearly underscores the importance of alleviating the state-space explosion issue.

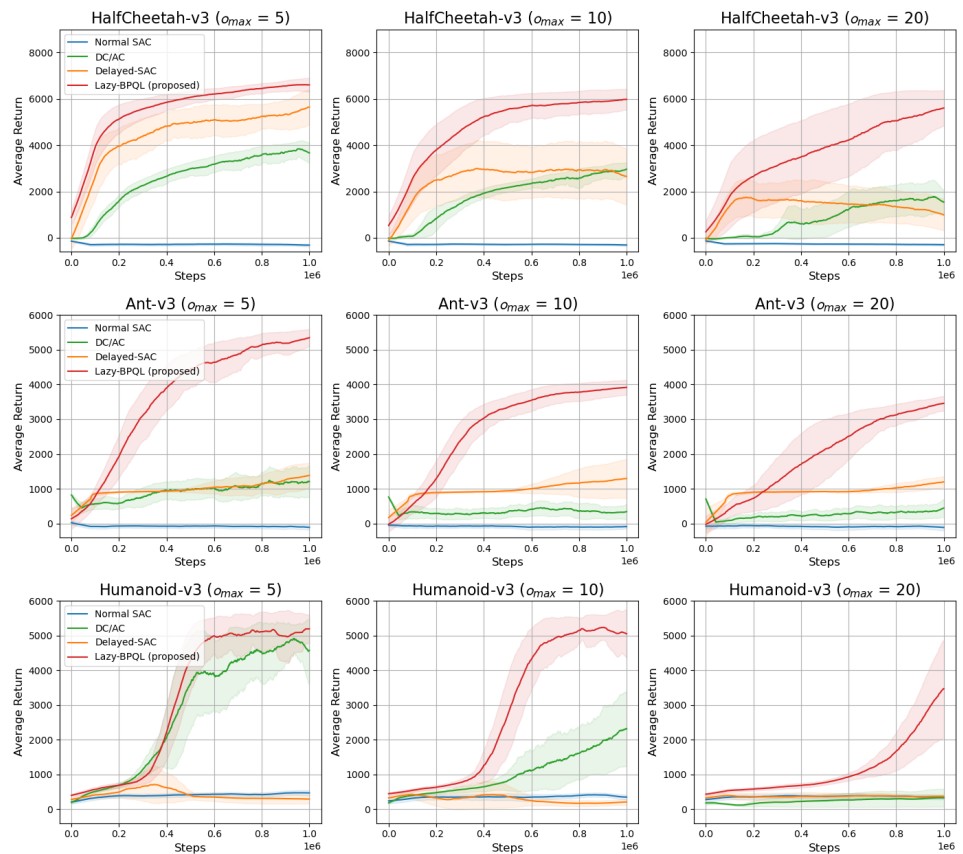

Figure 3: Performance curves of each algorithm on continuous control tasks in the MuJoCo benchmark with random delays of $o_{\max} \in \{5, 10, 20\}$. All tasks were conducted with five different seeds for one million time-steps. The shaded regions represent the standard deviation of average returns across the trials. Across all tasks, the proposed lazy-BPQL exhibits remarkable performance, consistently outperforming other algorithms. Additional results are provided in Appendix B.1.

### 6.2.4 ENVIRONMENTS WITH HIGHER RANDOMNESS

We evaluated the performance of lazy-BPQL in random-delay environments with increased randomness ($o_{\max} \in \{25, 30\}$) to empirically assess its robustness to greater randomness compared to other baseline algorithms. In the experiment, we included the second-best performing baseline, delayed-SAC, along with lazy-augmented-SAC to evaluate how effectively BPQL can mitigate the state-space explosion problem in such environments. The experiments were conducted on HalfCheetah-v3 and Ant-v3 tasks, and the result are listed in Table 6 in Appendix C.2. The results confirmed that lazy-BPQL exhibited performance degradation, but still maintained the best performance despite the increased randomness in delays, whereas other baselines were unable to learn any useful policies.

### 6.2.5 IMPACTS OF PROCESSING STATES IN ORDER

We investigated the impact of the assumption that states are used in order by comparing the performance of agents trained with and without this assumption. The results in Appendix C.3 reveal that the order in which observed states are used for decision-making can significantly affect the performance and learning stability of RL agents, with a notable drop in performance in the unordered case. Furthermore, the performance degradation becomes more pronounced as the randomness of delays increases. These findings seem to originate from the fact that both augmentation-based and model-based approaches heavily rely on preserving and understanding cause-and-effect relationships to restore the violated Markovian property caused by delays.

## 7 CONCLUSION

We investigated environments with random observation delays and proposed a novel approach to establish a connection to environments with constant delays by introducing a simple agent called the lazy-agent. With the proposed lazy-agents, random-delay environments can be transformed into their equivalent constant-delay counterparts, facilitating the application of state-of-the-art constant-delay approaches to random-delay environments without any modifications. We employed lazy-agents within the belief projection-based $Q$-learning (BPQL) framework, referred to as lazy-BPQL, to train our agents in equivalent constant-delay environments while effectively mitigating the state-space explosion issue of the augmentation-based approach. The empirical results demonstrated that the proposed lazy-BPQL significantly outperformed other baseline algorithms in terms of asymptotic performance and sample efficiency in random-delay environments, which strongly supports the efficacy of our approach.

It would be meaningful to employ our lazy-agents in real-world dynamic systems that suffer from random delays, where conventional constant-delay approaches are inadequate. In the future, we will extend the proposed algorithm to real-world applications, such as robotic locomotion and manipulation, by accounting for randomly varying sensor and actuator delays. We believe that the lazy-agents will play a pivotal role in extending conventional RL methods to real-world dynamic systems.

## 8 LIMITATIONS

Despite its notable advantages in constructing equivalent constant-delay environments from the original random-delay environments, employing the lazy-agent may encounter difficulties associated with long delays. This is because the constant delays in the equivalent environments are aligned with the maximum delay in the original random-delay environments. Consequently, the model-based approach may require more carefully designed dynamics models, as accumulated errors in recursive one-step predictions could result in significant performance degradation. On the other hand, the augmentation-based approach may confront the inherent state-space explosion issue. To circumvent this issue, we adopted a strategy of training lazy-agents within the BPQL framework, which can effectively mitigate the state-space explosion issue. Alternatively, the use of recurrent models, such as GRU (Cho, 2014), can also be considered, as explored in Firoiu et al. (2018). However, the necessity of knowing the maximum delay raises another concern, which may be unrealistic in some environments. This remains a challenge to be addressed in future work.

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

# A    RELATED WORK

Delays are prevalent in real-world reinforcement learning (RL) applications, arising from various factors such as computational time. Previous research on handling such delays within the RL framework can be categorized into two main approaches: augmentation-based and model-based.

**Augmentation-based approach**    The augmentation-based approach is often preferred in delayed environments since it allows us to retrieve the violated Markovian property (Altman & Nain, 1992; Katsikopoulos & Engelbrecht, 2003) and offers advantages for training agents through conventional RL algorithms in such environments. Despite its notable advantages, the method of augmenting the state itself remains a challenge associated with sample complexity, commonly referred to as the state-space explosion issue or the curse of dimensionality, which results in learning inefficiency. To mitigate this issue, DC/AC (Bouteiller et al., 2020) introduces an off-policy multi-step value estimation combined with a partial trajectory resampling method, which greatly enhances sample efficiency and accelerates the learning process. BPQL (Kim et al., 2023) proposes another novel approach to overcome such difficulty by introducing alternative representations for augmentation-based values. These alternative values are evaluated with respect to the original state space rather than the augmented one, thereby inherently mitigating the state-space explosion issue and demonstrating remarkable performance. More recently, AD-RL (Wu et al., 2024) alleviates the performance degradation stemming from this issue by leveraging auxiliary tasks with shorter delays to learn tasks with relatively long delays, reducing sample complexity and achieving notable performance.

**Model-based approach**    The model-based approach, also known as the state estimation method, aims to restore the Markovian property using learned dynamics models from underlying delay-free environments (Walsh et al., 2009; Firoiu et al., 2018; Chen et al., 2021; Derman et al., 2021). For example, delayed-$Q$ (Derman et al., 2021) employs an approximate feed-forward model to explicitly predict unobserved states, which can be learned from transition samples collected in delay-free environments. Using this model, the agent recursively predicts unobserved states through one-step predictions repeated over delayed time-steps and selects actions based on the predicted states. Similarly, Firoiu et al. (2018) proposes an approach that utilizes recurrent neural networks (Cho, 2014) to model the dynamics. These approaches facilitate sample-efficient learning without being affected by the issues associated with the sample complexity posed by the augmented state. However, approximation errors in building dynamics models may induce accumulated prediction errors, leading to suboptimal performance. Furthermore, the presence of noise in observations can exacerbate inaccuracies in learning dynamics models.

While numerous methods have shown promise, most works focus on the unrealistic assumption of constant delays, exhibiting nonnegligible performance degradation when applied to environments with random delays. Building upon previously proposed state-of-the-art methods, this study makes its primary contribution by proposing a novel approach that enables the handling of constant delays and random delays in exactly the same manner. Specifically, we propose a method to construct equivalent constant-delay environments from the original random-delay environments by introducing a simple agent termed the lazy-agent. This approach offers valuable insight that there is no need to devise new methods for handling random delays, as the lazy-agent naturally facilitates the application of conventionally proposed state-of-the-art methods, originally designed for constant delays, to random-delay environments without any modifications.

# B EXPERIMENTAL RESULTS

## B.1 PERFORMANCE COMPARISON

**Performance curves** We present the performance curves of each algorithm on the MuJoCo tasks with random delays of $o_{\max} \in \{5, 10, 20\}$. All tasks were conducted with five different seeds for one million time-steps. The shaded regions represent the standard deviation of average returns. Empirical results demonstrate that lazy-BPQL exhibits remarkable performance across all evaluated tasks.

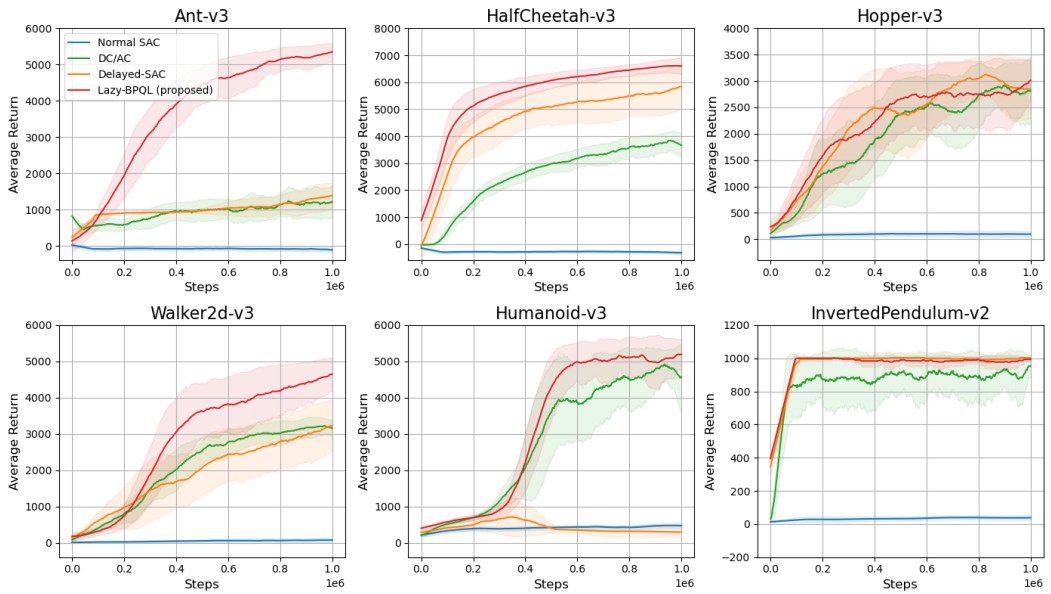

Figure 4: Performance curves of each algorithm on the MuJoCo tasks with $o_{\max} = 5$.

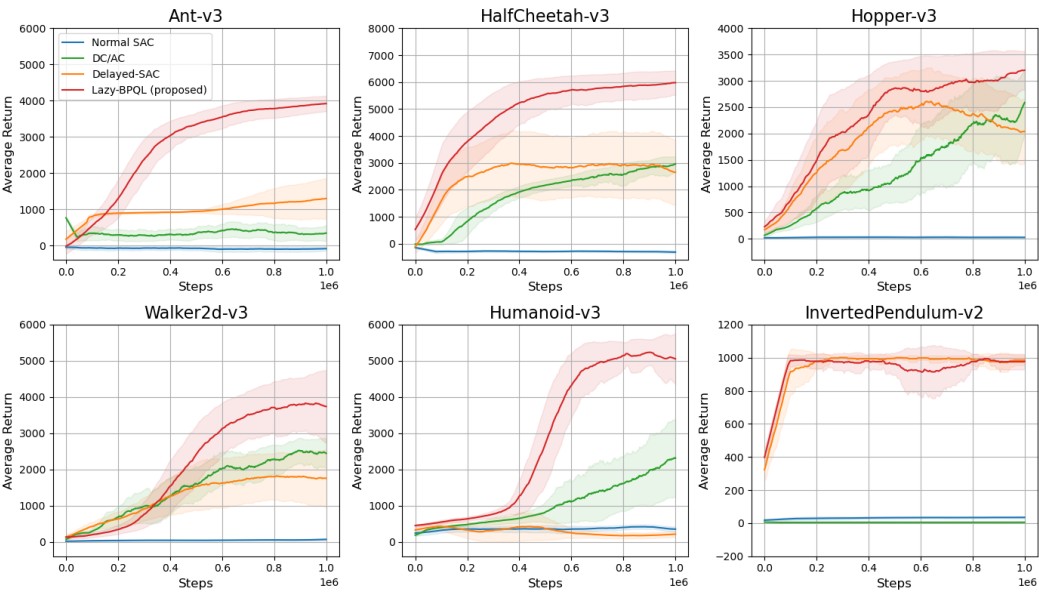

Figure 5: Performance curves of each algorithm on the MuJoCo tasks with $o_{\max} = 10$.

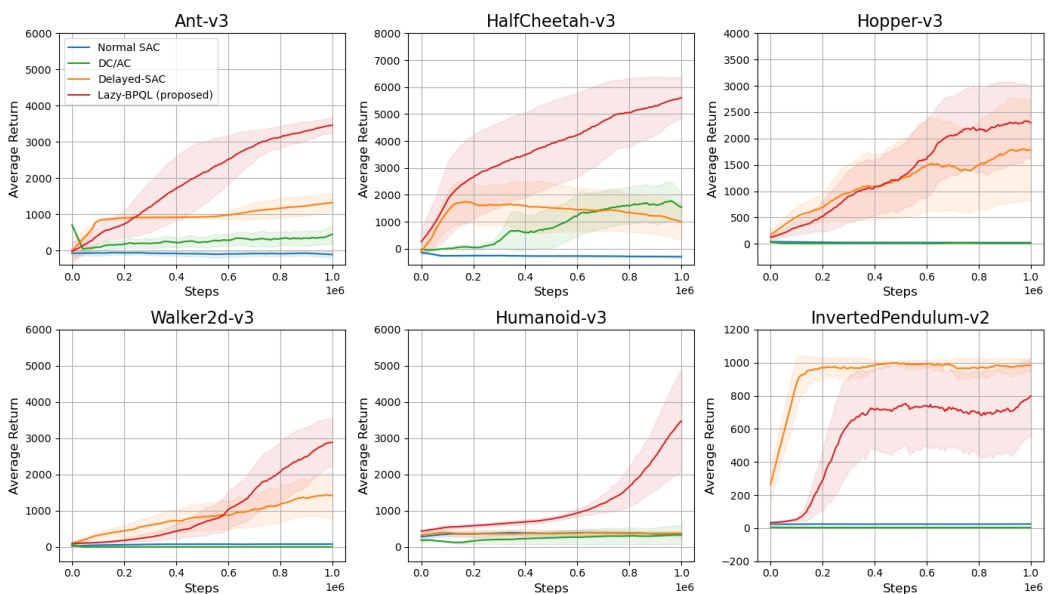

Figure 6: Performance curves of each algorithm on the MuJoCo tasks with $o_{\max} = 20$.

**Delay-free normalized scores**   To clarify the performance achieved by lazy-BPQL compared to other baseline algorithms, we report the delay-free normalized scores for each algorithm on the MuJoCo tasks in Fig. 7 and Table 2, following Wu et al. (2024). The delay-free normalized score is defined as $R_{\text{normalized}} = (R_{\text{algorithm}} - R_{\text{random}})/(R_{\text{delay-free}} - R_{\text{random}})$, where $R_{\text{algorithm}}$, $R_{\text{delay-free}}$, and $R_{\text{random}}$ represent the average returns of the baselines, delay-free SAC, and random policy, respectively. Here, delay-free SAC serves as the baseline performance in delay-free environments.

From the results, we confirmed that for tasks with $o_{\max} = \{5, 10\}$, lazy-BPQL exhibits the best performance comparable to the delay-free performance, achieving average scores of 0.91 and 0.81, respectively. It outperforms the second-best performing baselines by wide average margins of 0.28 and 0.34 points, each. Even for tasks with the longest maximum delay of $o_{\max} = 20$, lazy-BPQL maintains the highest average score. These scores further highlight the effectiveness of lazy-BPQL, demonstrating its superiority over other baseline algorithms across all evaluated tasks in MuJoCo.

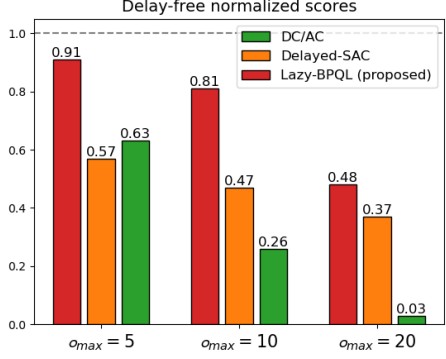

Figure 7: Delay-free normalized scores for each baseline algorithm, averaged across all the evaluated MuJoCo tasks. The dashed gray line represents the baseline score of delay-free SAC.

Table 2: Delay-free normalized scores of each algorithm with random delays of $o_{\max} \in \{5, 10, 20\}$. The best score is highlighted in **bold**.

| Environment | | Ant-v3 | HalfCheetah-v3 | Hopper-v3 | Walker2d-v3 | Humanoid-v3 | InvertedPendulum-v2 | Avg. |
|---|---|---|---|---|---|---|---|---|
| $o_{\max}$ | Algorithm | | | | | | | |
| | DC/AC | 0.28 | 0.32 | 0.79 | 0.62 | 0.86 | 0.88 | 0.63 |
| 5 | Delayed-SAC | 0.31 | 0.54 | 0.90 | 0.57 | 0.09 | **1.01** | 0.57 |
| | **Lazy-BPQL** (proposed) | **1.12** | **0.68** | **0.89** | **0.85** | **0.97** | 0.99 | **0.91** |
| | DC/AC | 0.11 | 0.23 | 0.51 | 0.45 | 0.29 | $-0.01$ | 0.26 |
| 10 | Delayed-SAC | 0.30 | 0.32 | 0.77 | 0.38 | 0.05 | **0.99** | 0.47 |
| | **Lazy-BPQL** (proposed) | **0.84** | **0.57** | **0.94** | **0.64** | **0.87** | 0.98 | **0.81** |
| | DC/AC | 0.09 | 0.12 | $-0.01$ | 0.00 | 0.04 | $-0.01$ | 0.03 |
| 20 | Delayed-SAC | 0.30 | 0.18 | 0.48 | 0.24 | 0.08 | **0.97** | 0.37 |
| | **Lazy-BPQL** (proposed) | **0.61** | **0.45** | **0.55** | **0.30** | **0.32** | 0.58 | **0.48** |

## B.2 EMPIRICAL RESULTS FOR PROPOSITION 5.1

To verify whether lazy-agents can perform in random-delay environments as if they were in constant-delay environments, we compared the performance of lazy-agents trained in **random-delay** environments (lazy-BPQL) with normal agents trained in **constant-delay** environments (BPQL) with constant delays set to $o = o_{\max}$. Each algorithm was evaluated for one million time-steps over five trials with different seeds on MuJoCo tasks, and the corresponding results are presented in Fig. 8, Table 3 and Table 4.

The results confirmed that both agents exhibited **almost identical** performance across all evaluated MuJoCo tasks with average margins of 0.1 in delay-free normalized scores (Wu et al., 2024). These empirical results strongly support our arguments that random-delay environments can be transformed into their equivalent constant-delay counterparts through the use of lazy-agents.

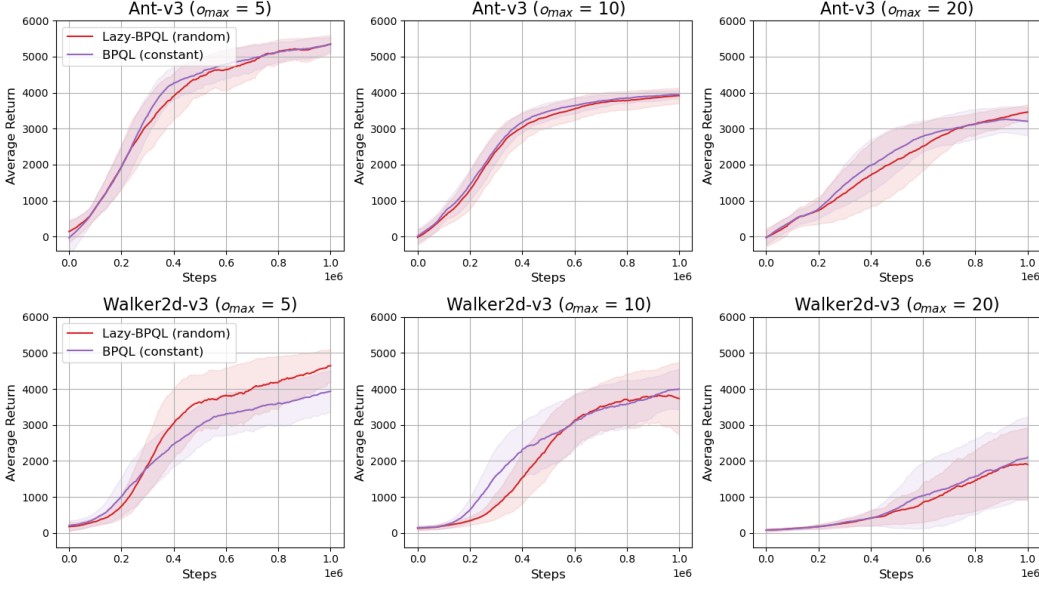

Figure 8: Performance curves of the proposed lazy-BPQL agents trained in random-delay environments with $o_{\max} \in \{5, 10, 20\}$ and the BPQL agent trained in constant-delay environments with $o = o_{\max}$ on continuous control tasks in the MuJoCo benchmark. All tasks were conducted with five different seeds for one million time-steps, and the shaded regions represent the standard deviation of average returns across the trials.

Table 3: Results of lazy-BPQL with random delays of $o_{\max} \in \{5, 10, 20\}$, and BPQL with constant delays of $o = o_{\max}$.

| | Environment | Ant-v3 | HalfCheetah-v3 | Hopper-v3 | Walker2d-v3 | Humanoid-v3 | InvertedPendulum-v2 |
|---|---|---|---|---|---|---|---|
| $o_{\max}$ | Algorithm | | | | | | |
| 5 | **Lazy-BPQL** (random-delay) | $3679.8_{\pm167}$ | $5583.9_{\pm169}$ | $2174.1_{\pm155}$ | $2843.2_{\pm272}$ | $3157.7_{\pm292}$ | $958.8_{\pm14}$ |
| | BPQL (constant-delay) | $3761.9_{\pm112}$ | $5212.7_{\pm41}$ | $2136.3_{\pm158}$ | $2577.4_{\pm157}$ | $3194.9_{\pm374}$ | $955.9_{\pm28}$ |
| 10 | **Lazy-BPQL** (random-delay) | $2744.5_{\pm112}$ | $4810.1_{\pm233}$ | $2300.9_{\pm164}$ | $2122.3_{\pm292}$ | $2820.5_{\pm348}$ | $936.9_{\pm38}$ |
| | BPQL (constant-delay) | $2831.9_{\pm103}$ | $4282.2_{\pm203}$ | $2129.2_{\pm184}$ | $2331.6_{\pm252}$ | $2891.5_{\pm357}$ | $934.7_{\pm20}$ |
| 20 | **Lazy-BPQL** (random-delay) | $1976.5_{\pm248}$ | $3727.2_{\pm279}$ | $1346.7_{\pm245}$ | $1025.7_{\pm302}$ | $1143.8_{\pm371}$ | $566.9_{\pm88}$ |
| | BPQL (constant-delay) | $2078.9_{\pm157}$ | $3062.7_{\pm252}$ | $1526.7_{\pm227}$ | $846.7_{\pm443}$ | $1197.7_{\pm457}$ | $608.7_{\pm210}$ |

Table 4: Delay-free normalized scores of lazy-BPQL with random delays of $o_{\max} \in \{5, 10, 20\}$, and BPQL with constant delays of $o = o_{\max}$.

| | Environment | Ant-v3 | HalfCheetah-v3 | Hopper-v3 | Walker2d-v3 | Humanoid-v3 | InvertedPendulum-v2 | **Avg.** | **Residue.** |
|---|---|---|---|---|---|---|---|---|---|
| $o_{\max}$ | Algorithm | | | | | | | | |
| 5 | **Lazy-BPQL** (random-delay) | 1.12 | 0.68 | 0.89 | 0.85 | 0.97 | 0.99 | 0.91 | 0.01 |
| | BPQL (constant-delay) | 1.14 | 0.62 | 0.88 | 0.74 | 0.99 | 1.00 | 0.90 | |
| 10 | **Lazy-BPQL** (random-delay) | 0.84 | 0.57 | 0.94 | 0.64 | 0.87 | 0.98 | 0.81 | 0.01 |
| | BPQL (constant-delay) | 0.86 | 0.51 | 0.88 | 0.70 | 0.89 | 0.97 | 0.80 | |
| 20 | **Lazy-BPQL** (random-delay) | 0.61 | 0.45 | 0.55 | 0.30 | 0.32 | 0.58 | 0.48 | 0.01 |
| | BPQL (constant-delay) | 0.64 | 0.35 | 0.63 | 0.25 | 0.34 | 0.58 | 0.47 | |

# C ABLATION STUDY

## C.1 STATE-SPACE EXPLOSION ISSUE

In this section, we present the performance of lazy-augmented-SAC and lazy-BPQL on the MuJoCo tasks with random delays of $o_{\max} \in \{5, 10, 20\}$. As listed in Table. 5, lazy-BPQL outperformed lazy-augmented-SAC across all evaluated tasks. Note that lazy-augmented-SAC completely failed to learn any useful policy even for tasks with $o_{\max} = 5$. These results clearly highlights the importance of mitigating the state-space explosion issue when employing augmentation-based approaches.

Table 5: Results of lazy-augmented-SAC and lazy-BPQL with random delays of $o_{\max} \in \{5, 10, 20\}$. Each algorithm was evaluated for one million time-steps over five trials with different seeds.

| | $o_{\max}$ | 5 | 10 | 20 |
|---|---|---|---|---|
| **Environment** | Algorithm | | | |
| Ant-v3 | **Lazy-BPQL** (proposed) | $\mathbf{3679.8}_{\pm167}$ | $\mathbf{2744.5}_{\pm112}$ | $\mathbf{1976.5}_{\pm248}$ |
| | Lazy-augmented-SAC | $898.5_{\pm93}$ | $913.2_{\pm29}$ | $721.4_{\pm86}$ |
| HalfCheetah-v3 | **Lazy-BPQL** (proposed) | $\mathbf{5583.9}_{\pm169}$ | $\mathbf{4810.1}_{\pm233}$ | $\mathbf{3727.2}_{\pm279}$ |
| | Lazy-augmented-SAC | $2137.2_{\pm361}$ | $1068.3_{\pm122}$ | $500.9_{\pm137}$ |

## C.2 ENVIRONMENTS WITH HIGHER RANDOMNESS

In this section, we provide the performance of lazy-BPQL in random-delay environments with increased randomness ($o_{\max} \in \{25, 30\}$) to empirically assess its robustness to greater randomness compared to other baseline algorithms. In experiment, we included the second-best performing baseline, delayed-SAC, along with lazy-augmented-SAC to verify how effectively BPQL can address the state-space explosion issue. The experiments were conducted in HalfCheetah-v3 and Ant-v3 tasks. Each algorithm was evaluated for one million time-steps over five trials with different seeds, and the results are listed in Table 6 and Table 7.

The results confirm that lazy-BPQL exhibited performance degradation, but still maintained the best performance despite the increased randomness in delays up to $o_{\max} = 30$, whereas other baselines were unable to learn any useful policies.

Table 6: Results of each baseline with random delays of $o_{\max} \in \{20, 25, 30\}$.

| $o_{\max}$ | | 20 | 25 | 30 |
|---|---|---|---|---|
| **Environment** | Algorithm | | | |
| Ant-v3 | **Lazy-BPQL** (proposed) | $\mathbf{1976.5}_{\pm 248}$ | $\mathbf{1944.3}_{\pm 176}$ | $\mathbf{1600.2}_{\pm 161}$ |
| | Lazy-augmented-SAC | $721.4_{\pm 86}$ | $466.3_{\pm 114}$ | $-34.3_{\pm 81}$ |
| | Delayed-SAC | $955.7_{\pm 110}$ | $949.9_{\pm 141}$ | $961.2_{\pm 154}$ |
| HalfCheetah-v3 | **Lazy-BPQL** (proposed) | $\mathbf{3727.2}_{\pm 279}$ | $\mathbf{2492.1}_{\pm 379}$ | $\mathbf{1971.1}_{\pm 265}$ |
| | Lazy-augmented-SAC | $500.9_{\pm 137}$ | $-5.8_{\pm 131}$ | $-199.1_{\pm 25}$ |
| | Delayed-SAC | $1377.8_{\pm 140}$ | $1076.5_{\pm 123}$ | $1194.8_{\pm 73}$ |

Table 7: Delay-free normalized scores of each baseline with random delays of $o_{\max} \in \{20, 25, 30\}$.

| $o_{\max}$ | | 20 | 25 | 30 |
|---|---|---|---|---|
| **Environment** | Algorithm | | | |
| Ant-v3 | **Lazy-BPQL** (proposed) | **0.61** | **0.60** | **0.49** |
| | Lazy-augmented-SAC | 0.23 | 0.15 | 0.07 |
| | Delayed-SAC | 0.30 | 0.29 | 0.31 |
| HalfCheetah-v3 | **Lazy-BPQL** (proposed) | **0.45** | **0.31** | **0.25** |
| | Lazy-augmented-SAC | 0.08 | 0.03 | 0.01 |
| | Delayed-SAC | 0.19 | 0.15 | 0.16 |

## C.3 IMPACT OF PROCESSING STATES IN ORDER

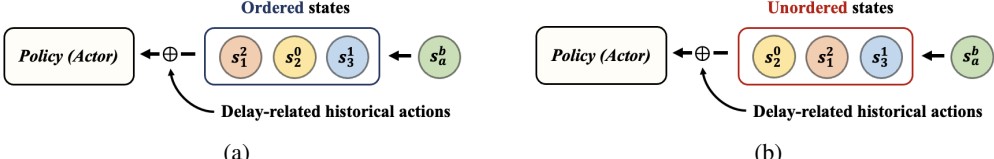

(a)                    (b)

Figure 9: The visual examples illustrating cases where (a) the observed states are processed in order and (b) the observed states are processed out of order. $\oplus$ denotes the concatenation operation.

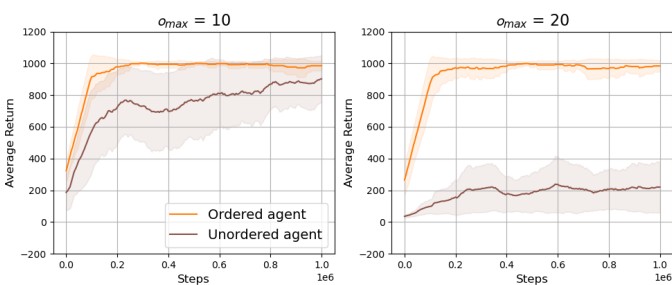

Figure 10: Results of ordered and unordered agent in InvertedPendulum-v2 MuJoCo task with random delays of $o_{\max} \in \{10, 20\}$.

In the presence of randomness in observation delays, states may be observed simultaneously, and their order can even become scrambled. When utilizing these scrambled states for decision-making in random-delay environments, they can be used either in the observed order (unordered state processing) or in their original generated order (ordered state processing).

We investigated the impact of the assumption that states are used in order by comparing the performance of agents trained with and without this assumption (see Fig. 9). In the experiment, we utilized delayed-SAC for learning InvertedPendulum-v2 task in MuJoCo, as it demonstrated respectable and stable performance in relatively simple tasks. We aimed to verify how this assumption impacts such performance, even in such simple task. We refer to the delayed-SAC agent trained in an ordered manner as the *ordered agent*, and the agent trained in a disordered manner as the *unordered agent*. Each agent was evaluated for one million time-steps over five trials with random seeds, and the corresponding results are presented in Fig. 10 and Table 8.

Table 8: Results of ordered and unordered agent in InvertedPendulum-v2 MuJoCo task with random delays of $o_{\max} \in \{10, 20\}$. The standard deviations of average returns are denoted by $\pm$.

| Environment | | InvertedPendulum-v2 |
|---|---|---|
| $o_{\max}$ | Algorithm | |
| 10 | Unordered agent | $739.5_{\pm 36}$ |
| | Ordered agent | $\mathbf{947.6}_{\pm 36}$ |
| 20 | Unordered agent | $181.6_{\pm 40}$ |
| | Ordered agent | $\mathbf{933.5}_{\pm 33}$ |

The results reveal that the order in which observed states are used can significantly affect the performance and learning stability of RL agents, with a notable drop in performance in the unordered case. Furthermore, the performance degradation becomes more pronounced as the randomness of delays increases. These findings seem to originate from the fact that both augmentation-based and model-based approaches heavily rely on preserving and understanding cause-and-effect relationships to restore the violated Markovian property caused by delays.

# D EXPERIMENTAL DETAILS

## D.1 ENVIRONMENTAL DETAILS

Table 9: Environmental details of the MuJoCo benchmark.

| Task | State dimension | Action dimension | Time-step (s) |
|---|---|---|---|
| Ant-v3 | 27 | 8 | 0.05 |
| HalfCheetah-v3 | 17 | 6 | 0.05 |
| Walker2d-v3 | 17 | 6 | 0.008 |
| Hopper-v3 | 11 | 3 | 0.008 |
| Humanoid-v3 | 376 | 17 | 0.015 |
| InvertedPendulum-v2 | 4 | 1 | 0.04 |

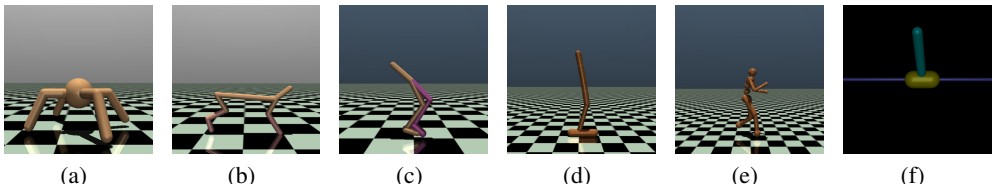

|       (a)       |       (b)       |       (c)       |       (d)       |       (e)       |       (f)       |

Figure 11: Experimental environments in the MuJoCo benchmark: (a) Ant-v3 (b) HalfCheetah-v3, (c) Walker2d-v3, (d) Hopper-v3, (e) Humanoid-v3, and (f) InvertedPendulum-v2

## D.2 IMPLEMENTATION DETAILS

The implementation details of the proposed lazy-BPQL align with those presented in Kim et al. (2023), with the specific hyperparameters listed in Table 10. Since the baseline algorithms included in our experiments employ the SAC algorithm as their foundational learning algorithm, the hyperparameters are consistent across all approaches, except for the DC/AC algorithm.

Table 10: Hyperparameters for lazy-BPQL and the baselines.

| Hyperparameters | Values |
|---|---|
| Actor network | 256, 256 |
| Critic network | 256, 256 |
| Learning rate (actor) | 3e-4 |
| Learning rate (critic) | 3e-4 |
| Temperature ($\alpha$) | 0.2 |
| Discount factor ($\gamma$) | 0.99 |
| Replay buffer size | 1e6 |
| Mini-Batch size | 256 |
| Target entropy | $-\dim|\mathcal{A}|$ |
| Target smoothing coefficient ($\xi$) | 0.995 |
| Optimizer | Adam (Kingma, 2014) |
| Total time-steps | 1e6 |

### D.3 Pseudo code of Lazy-BPQL

The proposed lazy-agent can be seamlessly integrated into the BPQL framework with minimal modifications by *using* the initial state for decision-making at its maximum delayed times. Subsequently, all states become naturally available for use at their respective maximum delayed times.

In the implementation, a temporary buffer $\mathcal{B}$ has been employed, as utilized by Kim et al. (2023), to store *observed* states, corresponding rewards, and action histories, which enables the agent to access timely and relevant information for constructing augmented states. Additionally, we have assumed that all feedback, including reward, is maximally delayed in equivalent constant-delay environments, similar to (Kim et al., 2023). Thus, the reward corresponding to the action $a_t$ is assumed to be $r_{t-o_{\max}}$.

---

**Algorithm 1** Lazy Belief Projection-based $Q$-Learning (Lazy-BPQL)

---

1: **Input:** actor $\bar{\pi}_\phi(a|\hat{x})$, beta critic $Q_{\theta,\beta}(s,a)$, target beta critic $Q_{\bar{\theta},\beta}(s,a)$, replay buffer $\mathcal{D}$, temporary buffer $\mathcal{B}$, maximum delay $o_{\max}$, beta critic learning rate $\lambda_Q$, actor learning rate $\lambda_{\bar{\pi}}$, soft update rate $\xi$, episodic length $H$, and total number of episodes $E$.
2: **for** episode $e = 1$ to $E$ **do**
3:     **for** time-step $t = 1$ to $H$ **do**
4:         **if** $t < o_{\max}$ **then**
5:             select random or 'no-ops' action $a_t$
6:             execute $a_t$ on environment
7:             put $a_t$, observed states, rewards to $\mathcal{B}$
8:         **else if** $t = o_{\max}$ **then**         $\triangleright$ wait for $o_{\max}$ time-steps
9:             select random or 'no-ops' action $a_t$
10:            execute $a_t$ on environment
11:            put $a_t$, observed states, rewards to $\mathcal{B}$
12:         **else**
13:            get $s_{t-o_{\max}}, a_{t-o_{\max}}, ..., a_{t-1}$ from $\mathcal{B}$
14:                    $\triangleright$ get most recent *usable* state and action histories
15:            $\hat{x}_t \leftarrow \left(s_{t-o_{\max}}, a_{t-o_{\max}}, ..., a_{t-1}\right)$     $\triangleright$ construct augmented state
16:            $a_t \leftarrow \bar{\pi}_\phi(\hat{x}_t)$
17:            execute $a_t$ on environment
18:            put $a_t$, observed states, rewards to $\mathcal{B}$
19:            **if** $t > 2o_{\max}$ **then**
20:                get $s_{t-2o_{\max}}, s_{t-2o_{\max}+1}, s_{t-o_{\max}}, r_{t-o_{\max}}, a_{t-2o_{\max}}, ..., a_{t-o_{\max}}$ from $\mathcal{B}$
21:                $\hat{x}_{t-o_{\max}} \leftarrow \left(s_{t-2o_{\max}}, a_{t-2o_{\max}}, ..., a_{t-o_{\max}}\right)$
22:                $\hat{x}_{t-o_{\max}+1} \leftarrow \left(s_{t-2o_{\max}+1}, a_{t-2o_{\max}+1}, ..., a_{t-o_{\max}+1}\right)$
23:                store $\left(\hat{x}_{t-o_{\max}}, s_{t-o_{\max}}, a_{t-o_{\max}}, r_{t-o_{\max}}, \hat{x}_{t-o_{\max}+1}, s_{t-o_{\max}+1}\right)$ in $\mathcal{D}$
24:                pop $s_{t-2o_{\max}}, a_{t-2o_{\max}}$ from $\mathcal{B}$
25:            **end if**
26:         **end if**
27:     **end for**
28:     **for** each gradient step **do**
29:         $\theta \leftarrow \theta - \lambda_Q \nabla \mathcal{J}_{Q_\beta}(\theta)$         $\triangleright$ update beta critic
30:         $\phi \leftarrow \phi - \lambda_{\bar{\pi}} \nabla \mathcal{J}_{\bar{\pi}}(\phi)$         $\triangleright$ update actor
31:         $\bar{\theta} \leftarrow \xi\theta + (1-\xi)\bar{\theta}$         $\triangleright$ update target beta critic
32:     **end for**
33: **end for**
34: **Output:** actor $\bar{\pi}_\phi$

---

As discussed in Section 3.1, the augmented reward for the action with respect to the augmented state is a random variable that has to be determined based on the conditional expectation as in equation 6. Fortunately, this expected value can be empirically obtained through the use of replay buffer $\mathcal{D}$:

$$\bar{r}_{t-o_{\max}} = \mathbb{E}_{(s,a)\sim\mathcal{D}}[r_{t-o_{\max}}] \tag{17}$$

where $\bar{r}_{t-o_{\max}}$ and $r_{t-o_{\max}}$ represent $\bar{R}(\hat{x}_{t-o_{\max}}, a_{t-o_{\max}})$ and $R(s = s_{t-o_{\max}}, a = a_{t-o_{\max}})$, each. Consequently, training the beta-critic and actor requires only the following set of experience tuples:

$$\left(\hat{x}_{t-o_{\max}}, s_{t-o_{\max}}, a_{t-o_{\max}}, r_{t-o_{\max}}, \hat{x}_{t-o_{\max}+1}, s_{t-o_{\max}+1}\right). \tag{18}$$

# E    VISUAL REPRESENTATION OF LAZY-AGENT

In this section, we provide a visual representation of the proposed lazy-agent employed in RDMDPs, where the maximum delay is set to $o_{\max} = 3$.

$*s_a^b : a$ = generated time, $b$ = delay

| Times | t = 1 | t = 2 | t = 3 | t = 4 | t = 5 | t = 6 | t = 7 | t = 8 | t = 9 | t = 10 |
|---|---|---|---|---|---|---|---|---|---|---|
| Generated states | | | | | | | | | | |
| Observed states | | | | | | | | | | |
| Usable states | | | | | | | | | | |
| Augmented states | | | | | | | | | | |
| Actions | | | | | | | | | | |

(a) Time $t = 0$

$*s_a^b : a$ = generated time, $b$ = delay

| Times | t = 1 | t = 2 | t = 3 | t = 4 | t = 5 | t = 6 | t = 7 | t = 8 | t = 9 | t = 10 |
|---|---|---|---|---|---|---|---|---|---|---|
| Generated states | $s_1^2$ | | | | | | | | | |
| Observed states | | | | | | | | | | |
| Usable states | | | | | | | | | | |
| Augmented states | 'no-ops' | | | | | | | | | |
| Actions | $a_1$ | | | | | | | | | |

(b) Time $t = 1$

$*s_a^b : a$ = generated time, $b$ = delay

| Times | t = 1 | t = 2 | t = 3 | t = 4 | t = 5 | t = 6 | t = 7 | t = 8 | t = 9 | t = 10 |
|---|---|---|---|---|---|---|---|---|---|---|
| Generated states | $s_1^2$ | $s_2^1$ | | | | | | | | |
| Observed states | | | | | | | | | | |
| Usable states | | | | | | | | | | |
| Augmented states | | 'no-ops' | | | | | | | | |
| Actions | $a_1$ | $a_2$ | | | | | | | | |

(c) Time $t = 2$

Figure 12: At times 1 and 2, the states $s_1^2$ and $s_2^1$ are generated but remain unobserved by the lazy-agent due to delays. In this scenario, the lazy-agent does nothing ('no-ops') until the initial state $s_1^2$ becomes usable.

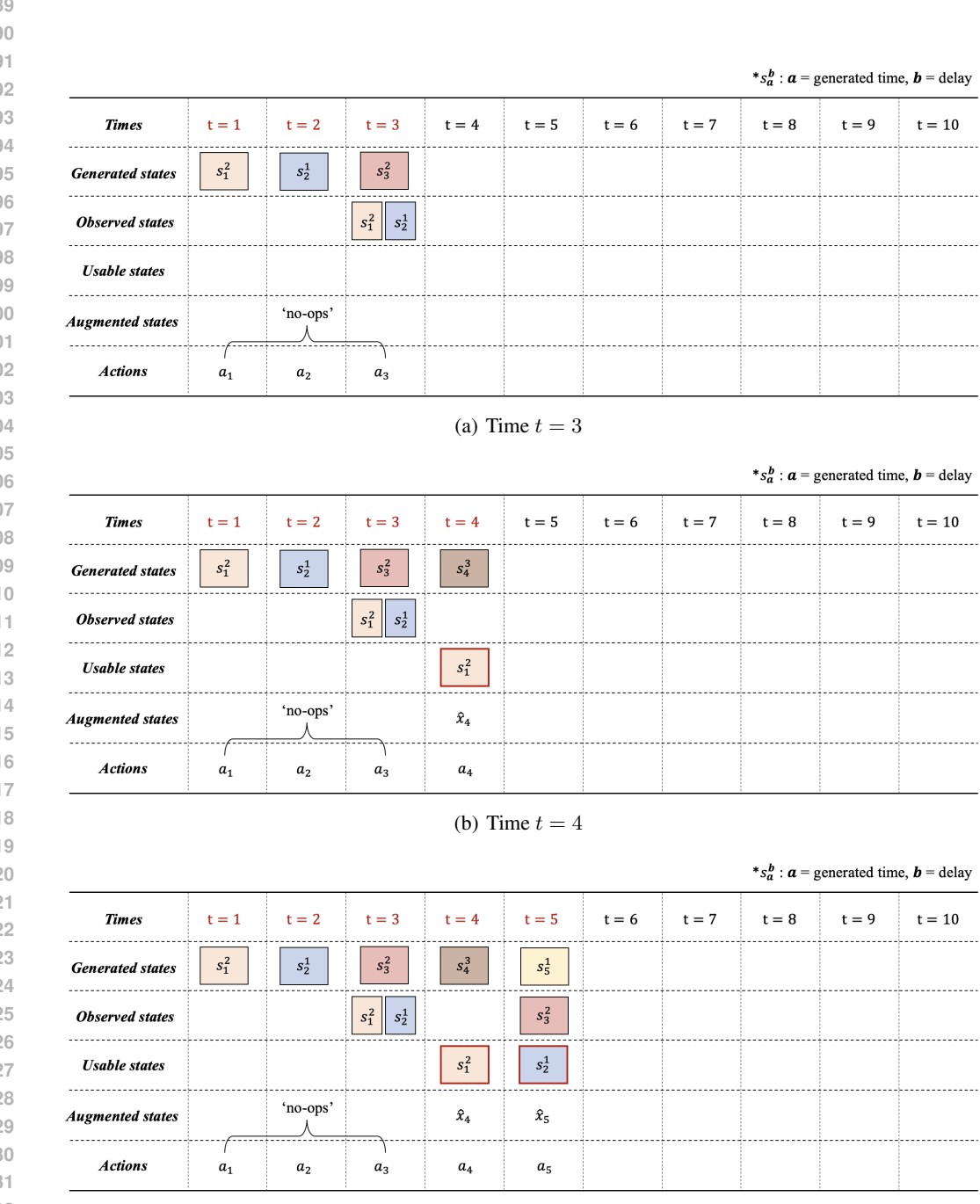

(a) Time $t = 3$

(b) Time $t = 4$

(c) Time $t = 5$

Figure 13: At time 3, states $s_1^2$ and $s_2^1$ are observed simultaneously. As the lazy-agent uses these observed states at their maximum delayed times, $s_1^2$ is used at time 4 and $s_2^1$ is used at time 5. These states are reformulated as augmented states before being fed into the policy, thereafter determining the appropriate actions. States $s_3^2$, $s_4^3$, and $s_5^1$ are generated at corresponding times, with $s_3^2$ being observed at time 5.

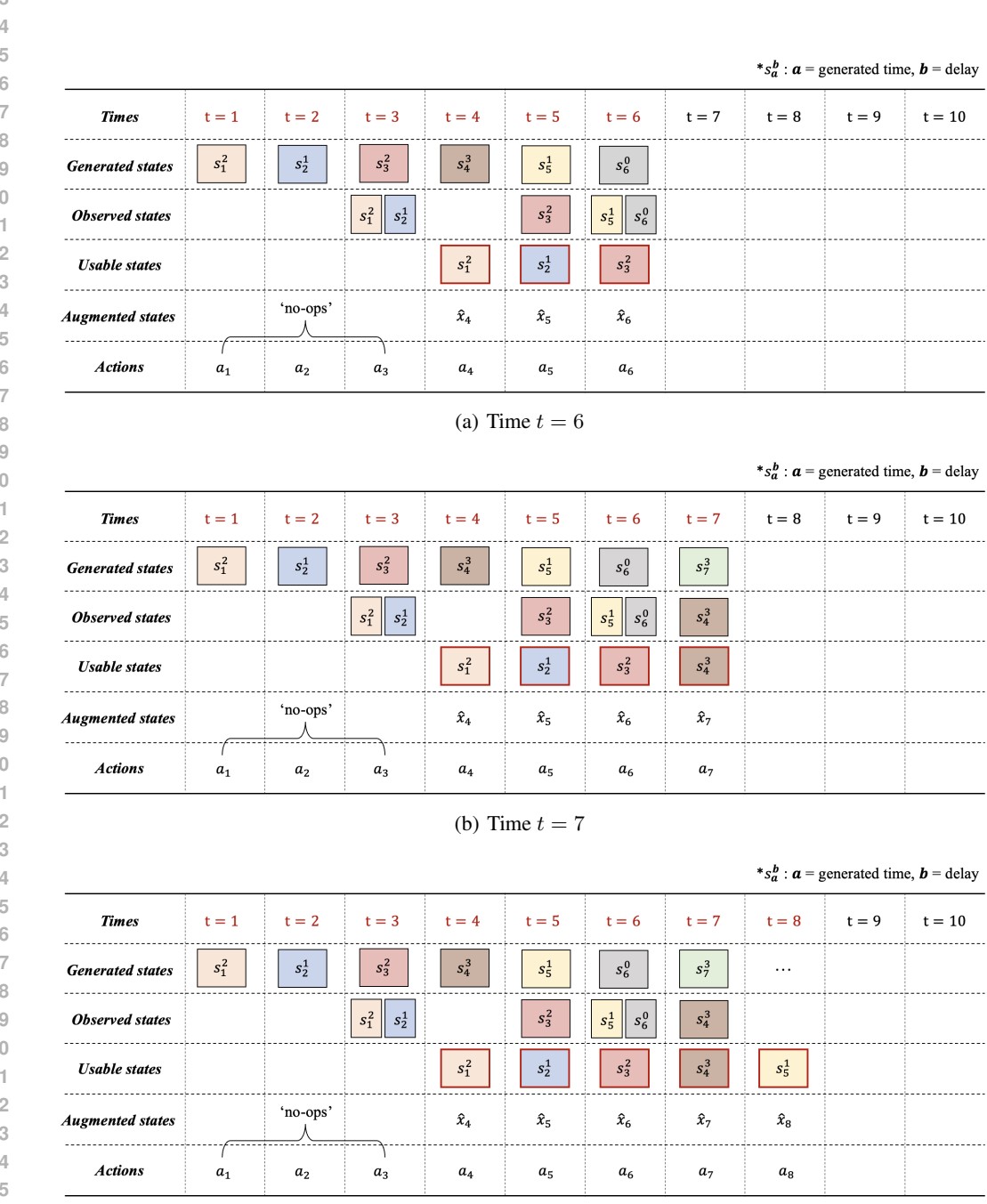

Figure 14: States $s_6^0$ and $s_7^3$ are generated at respective times. At time 6, states $s_5^1$ and $s_6^0$ are observed simultaneously but are not immediately usable because the previously generated states, $s_3^2$ and $s_4^3$, have not yet been used in decision-making processes. Instead, $s_3^2$ is used at this time. At time 7, state $s_4^3$ is observed and is available for use immediately. At time 8, state $s_5^1$ becomes usable, as all previously generated states have now been both observed and used.

$*s_a^b : a$ = generated time, $b$ = delay

| Times | t = 1 | t = 2 | t = 3 | t = 4 | t = 5 | t = 6 | t = 7 | t = 8 | t = 9 | t = 10 |
|---|---|---|---|---|---|---|---|---|---|---|
| Generated states | $s_1^2$ | $s_2^1$ | $s_3^2$ | $s_4^3$ | $s_5^1$ | $s_6^0$ | $s_7^3$ | ... | ... | |
| Observed states | | | $s_1^2$ $s_2^1$ | | $s_3^2$ | $s_5^1$ $s_6^0$ | $s_4^3$ | | | |
| Usable states | | | | $s_1^2$ | $s_2^1$ | $s_3^2$ | $s_4^3$ | $s_5^1$ | $s_6^0$ | |
| Augmented states | | 'no-ops' | | $\hat{x}_4$ | $\hat{x}_5$ | $\hat{x}_6$ | $\hat{x}_7$ | $\hat{x}_8$ | $\hat{x}_9$ | |
| Actions | $a_1$ | $a_2$ | $a_3$ | $a_4$ | $a_5$ | $a_6$ | $a_7$ | $a_8$ | $a_9$ | |

(a) Time $t = 9$

$*s_a^b : a$ = generated time, $b$ = delay

| Times | t = 1 | t = 2 | t = 3 | t = 4 | t = 5 | t = 6 | t = 7 | t = 8 | t = 9 | t = 10 |
|---|---|---|---|---|---|---|---|---|---|---|
| Generated states | $s_1^2$ | $s_2^1$ | $s_3^2$ | $s_4^3$ | $s_5^1$ | $s_6^0$ | $s_7^3$ | ... | ... | ... |
| Observed states | | | $s_1^2$ $s_2^1$ | | $s_3^2$ | $s_5^1$ $s_6^0$ | $s_4^3$ | | | $s_7^3$ |
| Usable states | | | | $s_1^2$ | $s_2^1$ | $s_3^2$ | $s_4^3$ | $s_5^1$ | $s_6^0$ | $s_7^3$ |
| Augmented states | | 'no-ops' | | $\hat{x}_4$ | $\hat{x}_5$ | $\hat{x}_6$ | $\hat{x}_7$ | $\hat{x}_8$ | $\hat{x}_9$ | $\hat{x}_{10}$ |
| Actions | $a_1$ | $a_2$ | $a_3$ | $a_4$ | $a_5$ | $a_6$ | $a_7$ | $a_8$ | $a_9$ | $a_{10}$ |

(b) Time $t = 10$

Figure 15: At times 9 and 10, states $s_6^0$ and $s_7^3$ are used in sequence. Despite the state observations occurring simultaneously or being out of order, all the delayed states are consistently used in sequence at their maximum delayed times, i.e., $\tau(s_n^{o_n}) = n + o_{\max}, \forall n > 0$.

