# OpenReview forum: "Reinforcement Learning via Lazy-Agent for Environments with Random Delays"
_ICLR.cc/2025/Conference — Submitted to ICLR 2025_

### Official Review · Reviewer_mDVd · 2024-10-19

**Soundness:** 3
**Presentation:** 3
**Contribution:** 2
**Rating:** 5
**Confidence:** 3

**Summary:**

The paper introduces Lazy-Agent for handling reinforcement learning in environments with random delays.
Traditionally, state-of-the-art RL techniques assume constant delays, simplifying the learning process but failing to reflect real-world scenarios' complexity.
A Lazy-Agent operates under the assumption that all feedback, such as states or rewards, is delayed by the maximum possible time, even though the actual delay is random.
This method allows the transformation of random-delay environments into constant-delay ones, making it possible to apply conventional RL techniques designed for constant delays.
Rather than reacting immediately, the Lazy-Agent waits for the maximum delay before making decisions.
This simplifies the management of random and unpredictable delays by treating them as constant delays, thereby facilitating the use of existing reinforcement learning methods for delayed feedback.
The paper extends the belief projection-based Q-learning (BPQL) framework, introducing Lazy-BPQL.

**Strengths:**

- The authors demonstrate that using Lazy-Agent can convert random-delay Markov decision processes (RDMDPs) into constant-delay MDPs (CDMDPs). This transformation allows established constant-delay solutions to be applied to random-delay environments.
- The proposed method is simple and easy to implement.

**Weaknesses:**

- For certain tasks (such as Ant, Hopper, Humanoid, and Pendulum), the performance gains from using Lazy-BPQL are minimal.
The reported improvements in performance, when considering standard deviation, are negligible.
- There is no significant advantage of Lazy Agents over traditional methods in all benchmarks, which raises concerns about the broader applicability of the Lazy Agent model.

**Questions:**

See weaknesses

---

> ### Author Response · Authors · 2024-11-23
> **Response to Reviewer mDVd (Part 1/2)**
>
> Dear Reviewer mDVd,
>
> We sincerely appreciate the time and effort you have spent reviewing our paper and providing valuable feedback. We have carefully considered each of your comments and provide the corresponding details as follows:
>
> ---
>
> ## **Contributions and performance gains from Lazy-BPQL**
>
> > *''For certain tasks (such as Ant, Hopper, Humanoid, and Pendulum), the performance gains from using Lazy-BPQL are minimal. The reported improvements in performance, when considering standard deviation, are negligible. There is no significant advantage of Lazy Agents over traditional methods in all benchmarks, which raises concerns about the broader applicability of the Lazy Agent model.''*
>
> To begin, let us briefly summarize the main contributions we offer, and then highlight the effectiveness of lazy-agents by presenting the corresponding results:
>
> This study makes its primary contribution by proposing a novel strategy that enables the handling of constant delays and random delays in exactly the same manner. **The proposed lazy-agent facilitates the application of state-of-the-art constant-delays approaches to random-delay environments without any modifications.**
>
> Through experiments, we have confirmed following two empiricial findings:
>
> - **Lazy-BPQL achieved the most remarkable performance compared to conventional algorithms that operate directly in random-delay environments**
> - **Lazy-agents performed in random-delay environments as if they were in constant-delay environments**
>
> and we present the corresponding results in below:
>
> ### **1. Performance of lazy-BPQL**
>
> To clarify the performance achievement by lazy-BPQL, we report *delay-free normalized scores* (Wu et al., 2024) [1] for each baseline algorithm on the MuJoCo tasks. The delay-free normalized score is defined as $R_{\text{normalized}} = (R_{\text{algorithm}}-R_{\text{random}})/(R_{\text{delay-free}}-R_{\text{random}})$, where $R_{\text{algorithm}}$, $R_{\text{delay-free}}$, and $R_{\text{random}}$ represent the average returns of each baseline algorithm, *delay-free SAC*, and random policy, respectively. Here, **delay-free SAC serves as the baseline performance in delay-free environments.**
>
> We present the task-specific scores for the evaluated tasks with the best scores in **bold** in the table below:
>
> | $o_\text{max}$ | Algorithm              | Ant-v3 | HalfCheetah-v3 | Hopper-v3 | Walker2d-v3 | Humanoid-v3 | InvertedPendulum-v2 | Avg.  |
> |-----------------|-----------------------|--------|----------------|-----------|-------------|-------------|---------------------|-------|
> | **5**          | DC/AC                | 0.28   | 0.32           | 0.79      | 0.62        | 0.86        | 0.88                | 0.63  |
> |                 | Delayed-SAC          | 0.31   | 0.54           | 0.90      | 0.57        | 0.09        | 1.01                | 0.57  |
> |                 | **Lazy-BPQL** (proposed) | **1.12**| **0.68**      | **0.89** | **0.85**    | **0.97**    | 0.99                | **0.91** |
> | **10**         | DC/AC                | 0.11   | 0.23           | 0.51      | 0.45        | 0.29        | -0.01               | 0.26  |
> |                 | Delayed-SAC          | 0.30   | 0.32           | 0.77      | 0.38        | 0.05        | **0.99**            | 0.47  |
> |                 | **Lazy-BPQL** (proposed) | **0.84**| **0.57**      | **0.94** | **0.64**    | **0.87**    | 0.98                | **0.81** |
> | **20**         | DC/AC                | 0.09   | 0.12           | -0.01     | 0.00        | 0.04        | -0.01               | 0.03  |
> |                 | Delayed-SAC          | 0.30   | 0.18           | 0.48      | 0.24        | 0.08        | **0.97**            | 0.37  |
> |                 | **Lazy-BPQL** (proposed) | **0.61**| **0.45**      | **0.55** | **0.30**    | **0.32**    | 0.58                | **0.48** |
>
>
> These results confirm that **lazy-BPQL demonstrated performance most comparable to delay-free baseline (close to 1.0)**, compared to other baseline algorithms from relatively short delays to long delays, outperforming the second-best performing baselines by notable average margins.
>
> ---
>
> **Reference**
>
> [1] Wu, Qingyuan, et al. "Boosting reinforcement learning with strongly delayed feedback through auxiliary short delays." Forty-first International Conference on Machine Learning.

---

> ### Author Response · Authors · 2024-11-23
> **Response to Reviewer mDVd (Part 2/2)**
>
> ### **2. Performance comparison between the lazy-agent and the normal agent**
>
> To verify whether lazy-agents can indeed perform in random-delay environments as if they were in constant-delay environments, we compared the performance of **lazy-agents trained in random-delay environments (lazy-BPQL)** with **normal agents trained in constant-delay environments (BPQL)** with $o=o_\text{max}$.
>
> We present their performance in terms of delay-free normalized scores, which are listed in table below:
>
> | $o_\text{max}$ | Algorithm                  | Ant-v3 | HalfCheetah-v3 | Hopper-v3 | Walker2d-v3 | Humanoid-v3 | InvertedPendulum-v2 | Avg.  | **Residue** |
> |-----------------|---------------------------|--------|----------------|-----------|-------------|-------------|---------------------|-------|-------------|
> | **5**          | **Lazy-BPQL**  | 1.12   | 0.68           | 0.89      | 0.85        | 0.97        | 0.99                | 0.91  | $\textcolor{red}{0.01}$        |
> |                 | BPQL    | 1.14   | 0.62           | 0.88      | 0.74        | 0.99        | 1.00                | 0.90  |             |
> | **10**         | **Lazy-BPQL**  | 0.84   | 0.57           | 0.94      | 0.64        | 0.87        | 0.98                | 0.81  | $\textcolor{red}{0.01}$        |
> |                 | BPQL    | 0.86   | 0.51           | 0.88      | 0.70        | 0.89        | 0.97                | 0.80  |             |
> | **20**         | **Lazy-BPQL**  | 0.61   | 0.45           | 0.55      | 0.30        | 0.32        | 0.58                | 0.48  | $\textcolor{red}{0.01}$        |
> |                 | BPQL   | 0.64   | 0.35           | 0.63      | 0.25        | 0.34        | 0.58                | 0.47  |             |
>
> From the results, both agents exhibited **almost identical performance** across all evaluated tasks (with average margins of 0.1), which strongly support our arguments that random-delay environments can be transformed into their equivalent constant-delay counterparts through the lazy-agents.
>
> The more detailed results are included in Appendix **B.1** and **B.2** in the revised paper.
>
> ---
>
> In conclusion, we believe **these findings highlight the efficacy of employing lazy-agent in random-delay environments, suggesting that training agents in equivalent constant-delay environments can be more effective than directly training them in the original random-delay environments.** Furthermore, although the lazy-agent was employed within the augmentation-based approach (i.e., BPQL) in our study, conventional model-based approaches can also be extended to random-delay environments using the lazy-agent, underscoring its broader applicability across both primary approaches for delayed environments.

---

> > ### Comment · Reviewer_mDVd · 2024-11-25
> >
> > I thank the authors for answering my questions, I will increase my rating to 5.

---

> > > ### Author Response · Authors · 2024-11-26
> > > **RE: Official Comment by Reviewer mDVd**
> > >
> > > We sincerely appreciate the reviewer's time and effort to reviewing our manuscript, and providing valuable feedback, and for raising his/her score!

---

### Official Review · Reviewer_PXdK · 2024-10-31

**Soundness:** 3
**Presentation:** 3
**Contribution:** 2
**Rating:** 6
**Confidence:** 3

**Summary:**

This paper proposes a way to transform random delay environments into constant delay environments, by imposing the specified maximum delay onto all observed states, regardless of their actual delay. Experiments show that agents trained under this transformation achieve similar results to agents trained in constant delay environments.

**Strengths:**

1. Nice visual representations e.g. Figure 1 help with understanding
2. The paper proposes a straightforward way to apply an algorithm for constant-delay environments to environments with random delays
3. Convincing number of environments and comparison algorithms used in experiments

**Weaknesses:**

1. Still requires a maximum number of delay time steps to be specified, which may be unrealistic in some environments (e.g. if observations must be sent over a high-latency network, if the observations require lengthy processing). This would admittedly be a challenging setting to address.
2. Only addresses the issue of large augmented state dimensions if the specified max delay is small.
3. I found the explanation of BPQL a bit confusing, as I don’t see how it avoids state space explosion if you are still training a policy on the augmented state.

**Questions:**

1. Unsure why you need to wait for $t>2_{o_{max}}$ in line 19 of Algorithm 1, instead of just $t>o_{max+2}$, if the purpose is just to wait until you have the next state/augmented state?
2. In section 3.2 it is stated that the state dimension could become infinite in infinite-horizon MDPs, wouldn’t this only occur if $O_{max}$ was infinite?
3. Could you provide more details on how BPQL mitigates the state space explosion problem, particularly given that the policy is still trained on the augmented state?

---

> ### Author Response · Authors · 2024-11-22
> **Response to Reviewer PXdK (Part 1/2)**
>
> Dear Reviewer PXdK,
>
> We would like to express our sincere gratitude for your in-depth review and valuable feedback, which has highlighted sections that might confuse readers. We are willing to do our best in order to address your concerns.
>
> ---
>
> ### **Required time-steps for constructing experience tuples.**
>
>
> > *''Unsure why you need to wait for $t > 2o_{\text{max}}$ in line 19 of Algorithm 1, instead of just $t > o_{\text{max}}+2$, if the purpose is just to wait until you have the next state/augmented state??''*
>
> In the implementation, we assumed that all feedback, including reward, is maximally delayed in equivalent constant-delay environments, similar to the BPQL framework. So, the reward corresponding to the action $a_t$ is considered to be $r_{t-{o_{\text{max}}}}$.
>
> Within the BPQL framework, the following set of experience tuples is stored in replay buffer $\mathcal{D}$ and utilized to train the beta-critic and actor:
>
> ($x_t, s_t, a_t, r_t, x_{t+1}, s_{t+1})$,
>
> where $x_{t} = (s_{t-{o_{\text{max}}}}, a_{t-{o_{\text{max}}}}, \cdots, a_{t-1})$ is the augmented state.
>
> Since we assumed that the state and reward are delayed by $o_{\text{max}}$ time-steps, the delayed state $s_{t-o_{\text{max}}}$ and delayed reward $r_{t-o_{\text{max}}}$ become available after time-step $t=o_{\text{max}}$. However, in order to construct the following experience tuple corresponding to the obtained delayed state and delayed reward,
>
>   $(x_{t-o_{\text{max}}}, s_{t-o_{\text{max}}}, a_{t-o_{\text{max}}}, r_{t-o_{\text{max}}}, x_{t-o_{\text{max}}+1}, s_{t-o_{\text{max}}+1})$,
>
> information delayed by up to $2o_{\text{max}}$ time-steps is further required to formulate the included augmented states. Specifically,
>
> $x_{t-o_{\text{max}}} = (s_{t-{{\color{red}2o_{\text{max}}}}}, a_{t-{{\color{red}2o_{\text{max}}}}}, \cdots, a_{t-o_{\text{max}}-1})$,
>
>
> $x_{t-o_{\text{max}}+1} = (s_{t-{{\color{red}2o_{\text{max}}}+1}}, a_{t-{{\color{red}2o_{\text{max}}}+1}}, \cdots, a_{t-o_{\text{max}}})$.
>
> Therefore, these augmented states can only be formulated after the time-step $t = 2o_{\text{max}}$.
>
> Consequently, the construction of the set of experience tuples for training beta-critic and actor can begin only after $t=2o_{\text{max}}$, rather than $t=o_{\text{max}}+2$.

---

> > ### Comment · Reviewer_PXdK · 2024-11-23
> >
> > Thank for for your response to my question.
> >
> > Unfortunately, I am still confused. I think I am confused about the change in notation from:
> >     $(x_t, s_t, a_t, r_t, x_{t+1}, s_{t+1})$
> >
> > To:
> >     $(x_{t-o_{max}}, s_{t-o_{max}}, a_{t-o_{max}}, r_{t-o_{max}}, x_{t-o_{max}+1}, s_{t-o_{max}+1})$
> >
> > In your Figure 13 b) (in Appendix E), the first augmented state is written as $x_4$ corresponding to $t=4$. Then in Figure Figure 13 c), once you have $s_5$ and $x_{t+5}$, I don't understand what the problem would be in constructing the $(x_4, s_4, a_4, r_4, x_{5}, s_{5})$ tuple? And this is only at $t=o_{max}+2$, not at $t=2o_{max}$.
> >
> > I hope this clarifies the issue, or possibly my misunderstanding.

---

> ### Author Response · Authors · 2024-11-22
> **Response to Reviewer PXdK (Part 2/2)**
>
> ### **Assumption of maximum bounded delays**
>
> > *''In section 3.2 it is stated that the state dimension could become infinite in infinite-horizon MDPs, would not this only occur if $o_{\text{max}}$ was infinite?''*
>
> We sincerely thank you for bringing this to our attention. As you pointed out, the maximum delay must be bounded to prevent the state dimension from becoming infinite. The method called *freeze* has also been proposed under the assumption of bounded maximum delays to address such cases in dealing with random delays using the augmented-based approach.
>
> Based on your valuable feedback, the following sentence in the original paper:
>
> >>*''This implies that its dimension will eventually reach infinity in infinite-horizon MDPs. To address this issue, a method called freeze has been proposed (Katsikopoulos & Engelbrecht, 2023).''*
>
> will be corrected and changed to:
>
> >>*''This implies that its dimension will eventually reach infinity in infinite-horizon MDPs without assuming a bounded maximum delay. Under the assumption of bounded maximum delays, Katsikopoulos & Engelbrecht (2023) proposed a method called freeze.''*
>
> Thank you for pointing this out. Your feedback has allowed us to improve the consistency of our paper. We will make the necessary modification in the revised version of our paper.
>
> ### **State-space explosion issue**
>
> > *''Could you provide more details on how BPQL mitigates the state space explosion problem, particularly given that the policy is still trained on the augmented state?''*
>
> As you pointed out, the augmented state is still required by the actor, unlike the critic, which takes the original state as input. The reason for this is as follows:
>
> During the training phase, we can compute the belief state (Liotet et al., 2022)[2] (which corresponds to the size of the original state space) from **previously collected** replay memory, allowing us to avoid the use of the augmented state for the critic. However, in the inference phase, the actor cannot access the belief state in real-time as the true observation does not reach. Therefore, to select optimal actions, the actor must perform inference using both delayed observations and the history of previous actions.
>
> Consider a game of ping pong. Making decisions solely based on delayed observations makes it difficult for the agent to select a proper action. Therefore, we should train the actor with the augmented states.
>
> Therefore, BPQL primarily addresses the state-space explosion problem from the perspective of learning the critics (i.e., beta-critics), while this issue may partially remain for the actors. Nevertheless, given its remarkable performance improvements over previously presented conventional constant-delay approaches, we believe BPQL remains a promising method with potential for further development, particularly regarding the actor. For this reason, we chose it as the baseline framework for training our lazy-agent.
>
> **Reference**
>
> [1] Katsikopoulos, Konstantinos V., and Sascha E. Engelbrecht. "Markov decision processes with delays and asynchronous cost collection." IEEE transactions on automatic control 48.4 (2003): 568-574.
>
>
> [2] Liotet, Pierre, et al. "Delayed reinforcement learning by imitation." International Conference on Machine Learning. PMLR, 2022.

---

> > ### Comment · Reviewer_PXdK · 2024-11-23
> >
> > Thank you for your responses to my questions, and for adding the additional explanation to the paper.
> >
> > Given your response to my second question, I feel that the language is a little strong throughout the paper e.g. "effectively avoiding the state-space explosion issue", when realistically the actor does still suffer from the issue. I agree that BPQL is a good baseline, particularly considering the Section 6.2.3 ablation, however the language confused me as I could not understand how the actor was avoiding the issue.

---

> > > ### Author Response · Authors · 2024-11-24
> > > **RE : Response to the Reviewer PXdK (Part 2/2)**
> > >
> > > > *''Given your response to my second question, I feel that the language is a little strong throughout the paper e.g. "effectively avoiding the state-space explosion issue", when realistically the actor does still suffer from the issue. I agree that BPQL is a good baseline, particularly considering the Section 6.2.3 ablation, however the language confused me as I could not understand how the actor was avoiding the issue.''*
> > >
> > > We sincerely thank you for your valuable comment regarding the use of language in our paper. We agree that terms such as "avoid" or "address" could be misleading, as the actor still partially suffers from the state-space explosion issue.
> > >
> > > To clarify this, we will revise the language in paper by replacing these terms with more appropriate ones, such as ''mitigate'' or ''alleviate''.
> > >
> > > We will ensure that this clarification is made in the revised paper, and we truly appreciate you for bringing this to our attention !

---

> ### Author Response · Authors · 2024-11-24
> **RE : Response to the Reviewer PXdK (Part 1/2)**
>
> Dear PXdK,
>
> We sincerely appreciate your engagement with our responses. We provide a detailed explanation in below:
>
> ---
>
> > *''In your Figure 13 b) (in Appendix E), the first augmented state is written as $x_4$ corresponding to $t=4$. Then in Figure 13 c), once you have $s_5$ and $x_{t+5}$, I don't understand what the problem would be in constructing the ($x_{4}$, $s_4$ , $a_4$, $r_4$, $x_5$, $s_5$) tuple? And this is only at $t = o_{\text{max}}+2$, not at $t = 2o_{\text{max}}$.''*
>
> As you pointed out, the augmented state $x_{4}$ can be formulated at time $t = 4 (=o_{\text{max}}+1)$, as the required state $s_1$ and historical actions ($a_1, a_2, a_3$) are available at this time (we assumed $o_{\text{max}}=3$).
>
> To be more specific, the augmented state $x_{4}$ is formulated as:
>
> $x_{4} = (s_{4-o_{\text{max}}}, a_{4-o_{\text{max}}}, a_{4-o_{\text{max}}+1}, a_{4-o_{\text{max}}+2}) = (s_1, a_1, a_2, a_3)$
>
> However, in constructing the following experience tuple:
>
> ($x_{4}$, $\textcolor{red}{s_4}$ , $a_4$, $r_4$, $x_5$, $s_5$),
>
> we still have no access to the state $\textcolor{red}{s_4}$ at time $t = 5(=o_{\text{max}}+2)$, since we assumed that all states are delayed by $o_{\text{max}}$ time-steps. Even if the state $\textcolor{red}{s_4}$ has already been observed at this time, it remains unused until the maximum delay time is reached, which occurs at $t = 7(=2o_{\text{max}}+1)$ (see $s^{3}_{4}$ described by the brown-colored box in Figure 13 (b)).
>
> This explains why we begin constructing experience tuples only after time $t = 2o_{\text{max}}$.
>
> We hope this response helps clarify any points of confusion you have.

---

> > ### Comment · Reviewer_PXdK · 2024-11-26
> >
> > Thank you very much for this explanation, this makes sense. Thank you also for addressing my issue with the language. I will raise my rating to a 6.

---

> ### Author Response · Authors · 2024-11-26
> **RE: Official Comment by Reviewer PXdK**
>
> We are truly glad to hear that it is now much clearer to you!
> If there are any further issues, please let us know, and we are willing to do our best to address them.
>
> Once again, we deeply appreciate your time and effort in providing constructive and valuable feedback on our manuscript, and for updating the score!

---

### Official Review · Reviewer_aMSF · 2024-11-04

**Soundness:** 2
**Presentation:** 3
**Contribution:** 2
**Rating:** 5
**Confidence:** 4

**Summary:**

This paper presents a derivation and a small-scale empirical analysis that shows random-delay MDPs can be transformed to nearly equivalent constant-delay MDPs by employing a lazy-agent that assumes that all states are (constantly) delayed by the maximum delayed times.

**Strengths:**

Soundness
======
 The approach is sound, and, albeit a small increment over existing approaches, well demonstrated empirically. The paper lacks an in-depth discussion of the benefits and limitations of the approach.

Significance & Related work
=========
There is no related work section in the paper and thus the significance of the work is not clarified.

Experimentation
=========
The experimental analysis shows the equivalence of RDMDPs to CDMDPs in BPQL in the MuJoCo environments.


Presentation
=========
The paper is well written.

**Weaknesses:**

Soundness
======
The paper lacks an in-depth discussion of the benefits and limitations of the approach.

Significance & Related work
=========
There is no related work section in the paper and thus the significance of the work is not clarified.

Experimentation
=========

There is a need, however, for more in-depth ablation experiments that evaluate the impact of assuming states are processed in order, and of highly varied random delays.

**Questions:**

* What is the performance of the approach in the other MuJoCo environments?

---

> ### Author Response · Authors · 2024-11-24
> **Response to Reviewer aMSF (Part 1/2)**
>
> Dear Reviewer aMSF,
>
> We sincerely appreciate your efforts in providing valuable feedback and expressing interests in our paper. We have carefully reviewed your comments and would like to provide the corresponding responses to the best of our ability.
>
> ---
>
> ## **Experimental results in other environments in MuJoCo**
>
> > *``What is the performance of the approach in the other MuJoCo environments?''*
>
> We evaluated the performance of the proposed lazy-BPQL in the following three additional environments with random delays of $o_{\text{max}} \in$ {5, 10, 20}:
>
> - HumanoidStandup-v4
> - Pusher-v4
> - Swimmer-v4
>
> and present the corresponding results below:
>
> ---
>
> ### **Average returns**
>
> In the experiment, we compared the performance of lazy-BPQL with the second-best performing baseline, delayed-SAC.
>
> The results are listed in table below, with the best performance highlighted in **bold**:
>
> | **$o_\text{max}$** | **Algorithm**         | **HumanoidStandup-v4**      | **Pusher-v4**           | **Swimmer-v4**          |
> |---------------------|-----------------------|-----------------------------|--------------------------|--------------------------|
> | $\times$           | Delay-free SAC        | $\color{blue}138491.7_{\pm8208}$         | $\color{blue}-27.9_{\pm3}$      | $\color{blue}203.9_{\pm99}$        |
> |                      | Random policy          | 33367.3$_{\pm667}$         | -149.6$_{\pm2}$       | 0.5$_{\pm2.5}$      |
> | 5                   | **Lazy-BPQL (proposed)** | **143143.8**$_{\pm14491}$         | **-30.7**$_{\pm3}$      | **254.4**$_{\pm64}$        |
> |                     | Delayed-SAC          | 115563.3$_{\pm27095}$         | -71.9$_{\pm14}$       |   66.9$_{\pm46}$  |
> | 10                  | **Lazy-BPQL (proposed)** | **149304.2**$_{\pm9000}$         | **-33.1**$_{\pm3}$      | **130.9**$_{\pm105}$        |
> |                     | Delayed-SAC          | 74168.5$_{\pm26653}$         | -68.4$_{\pm11}$       | 59.4$_{\pm33}$   |
> | 20                  | **Lazy-BPQL (proposed)** | **137188.4**$_{\pm10433}$         | **-38.1**$_{\pm3}$      | **140.8**$_{\pm73}$        |
> |                     | Delayed-SAC          | 61723.2$_{\pm16603}$         | -72.2$_{\pm12}$       | 41.2$_{\pm22}$ |
>
> Here, **delay-free SAC serves as the baseline performance in delay-free environments** (denoted by a blue-colored text).
>
> These results confirm that **lazy-BPQL outperformed delayed-SAC by significantly wide margins across all tasks, even exceeding the delay-free baseline performance in some cases.**
>
> ---
>
> ### **Delay-free normalized scores**
>
> To clarify the performance achievement by lazy-BPQL, we report *delay-free normalized scores* (Wu et al., 2024) [1], which is defined as $R_{\text{normalized}} = (R_{\text{algorithm}}-R_{\text{random}})/(R_{\text{delay-free}}-R_{\text{random}})$, where $R_{\text{algorithm}}$, $R_{\text{delay-free}}$, and $R_{\text{random}}$ represent the average returns of each baseline algorithm, delay-free SAC, and random policy, respectively.
>
> The task-specific scores are listed in table below, with the best scores highlighted in **bold**:
>
> | **$o_\text{max}$** | **Algorithm**         | **HumanoidStandup-v4**      | **Pusher-v4**           | **Swimmer-v4**          | **Avg.** |
> |---------------------|-----------------------|-----------------------------|--------------------------|--------------------------|--------------------------------|
> | 5                   | **Lazy-BPQL (proposed)** | **1.04**         | **0.97**      | **1.24**     | **1.08** |
> |                     | Delayed-SAC          | 0.78         | 0.63      |   0.32      | 0.57 |
> | 10                  | **Lazy-BPQL (proposed)** | **1.10**         | **0.95**      | **0.64**        | **0.89** |
> |                     | Delayed-SAC          | 0.39         | 0.66       |  0.28  | 0.44 |
> | 20                  | **Lazy-BPQL (proposed)** | **0.99**         | **0.91**      | **0.68** | **0.86** |
> |                     | Delayed-SAC          | 0.27          | 0.63       |  0.19 | 0.36 |
>
> These results further highlight the efficacy of the proposed lazy-BPQL, **demonstrating performance comparable to the delay-free baseline (close to 1.0)** and its robustness in maintaining performance even in environments with increased randomness in delays.
>
> ---
>
> ### **Environmental details**
>
> | **Environment**               | **State Dimension** | **Action Dimension** | **Time-step (s)** |
> |-------------------------|---------------------|-----------------------|--------------------|
> | HumanoidStandup-v4      | 376                 | 17                    | 0.05              |
> | Swimmer-v4              | 8                   | 2                     | 0.04              |
> | Pusher-v4               | 23                  | 7                     | 0.05              |
>
> ---
>
> **Reference**
>
> [1] Wu, Qingyuan, et al. "Boosting reinforcement learning with strongly delayed feedback through auxiliary short delays." Forty-first International Conference on Machine Learning

---

> ### Author Response · Authors · 2024-11-24
> **Response to Reviewer aMSF (Part 2/2)**
>
> Thank you for providing valuable and constructive feedback regarding the absence of discussions on limitations, related work, and additional experiments for a more in-depth exploration of the benefits of our study.
>
> We have carefully considered each of your comments and have included the corresponding details in the revised paper as follows:
>
> ---
>
> ## **Related works and benefits**
>
> We explored two primary approaches for handling delays within the RL framework: the augmentation-based approach and the model-based approach. The strengths and weaknesses of each approach are summarized to provide a comprehensive understanding of their benefits and limitations. Subsequently, we outlined the primary contribution of our study with respect to the aforementioned methods.
>
> **Details are included in Appendix A of the revised paper.**
>
> ---
>
> ## **Limitations**
>
> **The following limitations are discussed in Section 8 of the revised paper.**
>
> - The requirement of knowing the bounded maximum delay.
> - The alignment of constant delays in the equivalent constant-delay environments with maximum delays in random-delay environments.
>
> To alleviate the second issue, we employed a strategy of training lazy-agents within the BPQL framework (Kim et al., 2023) [2], which can effectively mitigate the state-space explosion issue caused by long delays. However, the necessity of knowing the maximum delay raises another concern, since it may be unrealistic in some real-world environments. This remains a challenge to be addressed in future work.
>
> ---
>
> ## **Additional experiments**
>
> We conducted the additional experiments:
>
> - Investigating the impact of processing states in order.
> - Evaluating the performance of the proposed lazy-BPQL in random-delay environments with higher randomness.
>
> We investigated the impact of processing states in order by comparing the performance of agents trained with ordered states and unordered states.
>
> We also evaluated the performance of lazy-BPQL in random-delay environments with increased randomness ($o_{\text{max}} \in$ {25, 30}) to empirically assess its robustness to higher randomness in delays, compared to other baseline algorithms.
>
> **Details for these experiments are provided in Section 6 and Appendix C of the revised paper.**
>
> ---
>
> **Reference**
>
> [2] Kim, Jangwon, et al. "Belief projection-based reinforcement learning for environments with delayed feedback." Advances in Neural Information Processing Systems 36 (2023): 678-696.
>
> ---
>
> Thank you once again for your valuable comments. Your feedback has greatly helped us improve the consistency and overall quality of our paper!

---

> > ### Comment · Reviewer_aMSF · 2024-11-27
> > **Thank you for the clarification and revision**
> >
> > I thank the authors for their comments and revision. I will retain my score.

---

> ### Author Response · Authors · 2024-11-27
> **RE: Thank you for the clarification and revision**
>
> We sincerely appreciate the time and effort you have dedicated to evaluating our work, and for providing valuable feedback.
>
> We hope that these comments and revisions sufficiently address your concerns. Thank you once again for your valuable and constructive feedback!

---

### Author Response · Authors · 2024-12-02
**General Response**

We would like to express our sincere gratitude to all the reviewers for taking the time to review our manuscript, and providing valuable feedback. The reviewers' comments have greatly contributed to enhancing the overall quality and clarity of our study.

We would like to : 1) summarize and highlight the key contributions of our study, and 2) outline the major modifications made to the revised manuscript in response to the reviewers' comments.

---

## **Contributions**

This study makes a primary contribution by introducing the *lazy-agent* that straightforwardly transforms random-delay environments into their equivalent constant-delay environments. This transformation enables the application of state-of-the-art constant-delay approaches to random-delay environments without any modifications.

Empirically, we demonstrate that the BPQL framework [1], originally designed to manage constant delays, achieves exceptional performance in random-delay environments when integrated with our lazy-agents, termed *lazy-BPQL*. This lazy-agent-based approach outperforms other baseline algorithms that directly operate in random-delay environments.

These results offer valuable insights that there is no need to devise new methods for handling random delays, as the lazy-agent naturally facilitates the application of conventional constant-delay approaches to random-delay environments.

---

## **Major Updates in the Revised Manuscript**

The followings contents have been updated or newly included in the revised manuscript:

- Experiments investigating 1) the impact of assuming ordered states and 2) the performance of lazy-BPQL in random-delay environments with higher randomness $o_{\text{max}} \in$ {25, 30} are included.
- A section for related works is included to provide a comprehensive understanding of conventional approaches and to highlight our contributions.
- A section for limitations is included to clarify the strengths and weaknesses of our study.
- To clarify the performance achieved by lazy-BPQL compared to other baseline algorithms, we include a delay-free baseline, *delay-free SAC*, and report delay-free normalized scores [2].
- Figure 3 from the original manuscript has been moved to Appendix B.2 in order to avoid any points of confusion and improve readability.

The modified parts are highlighted in red in the revised version of our manuscript.

---

Lastly, we would like to thank the reviewers once again for their efforts and thoughtful comments.

Best regards,

Authors

---

**Reference**

[1] Kim, Jangwon, et al. "Belief projection-based reinforcement learning for environments with delayed feedback." Advances in Neural Information Processing Systems 36 (2023): 678-696.

[2] Wu, Qingyuan, et al. "Boosting reinforcement learning with strongly delayed feedback through auxiliary short delays." Forty-first International Conference on Machine Learning

---

### Meta-Review · Area_Chair_MLrE · 2024-12-19

**Metareview:**

Reinforcement Learning via Lazy-Agent for Environments with Random Delays

Summary: The paper addresses the challenge of handling random delays in reinforcement learning (RL), which disrupt the Markovian property and complicate policy optimization. The authors propose a new framework utilizing a "lazy-agent" to transform random-delay environments into their constant-delay equivalents, enabling the application of existing RL algorithms without modifications. The approach uses a belief projection-based Q-learning (BPQL) framework to mitigate state-space explosion issues caused by delay variability. Empirical evaluations on MuJoCo benchmarks demonstrate that the lazy-agent approach outperforms existing methods.

Comments: We received three reviews, with the scores  5, 5, 6, and the average score is 5.33.

Reviewers have given positive comments about some aspects of the paper. In particular, the reviewers appreciated the conceptually simple and clear algorithm to address this challenging problem. The reviewers also mentioned that the paper is well-written. However, the reviewers have raised multiple concerns about the paper which outweigh the positive aspects. Reviewer aMSF has mentioned the lack of a clear literature review that makes it difficult to evaluate the original contributions of this paper. Reviewer PXdK has commented that the paper does not really avoid the state space explosion problem which the authors acknowledge in the rebuttal. Reviewer mDVd mentioned that for certain tasks, the performance gains from using Lazy-BPQL are minimal. Moreover, there is no significant advantage of Lazy Agents over traditional methods in all benchmarks.

Overall, while the lazy-agent framework is conceptually interesting and the paper is well-presented, it still has significance weaknesses as pointed out above. Addressing these weaknesses and improving the experimental and theoretical rigor would enhance the paper’s contribution.

**Additional Comments On Reviewer Discussion:**

Please see the "Comments" in the meta-review.

---

### Decision · Program_Chairs · 2025-01-22

Reject